# A new interpretative framework for below-cloud effects on stable water isotopes in vapour and rain

Pascal Graf[1], Heini Wernli[1], Stephan Pfahl[1,2], and Harald Sodemann[1,3,4]

[1]Institute for Atmospheric and Climate Science, ETH Zurich, Zurich, Switzerland
[2]Institute of Meteorology, Freie Universität Berlin, Berlin, Germany
[3]Geophysical Institute, University of Bergen, Bergen, Norway
[4]Bjerknes Centre for Climate Research, Bergen, Norway

**Correspondence:** Harald Sodemann (harald.sodemann@uib.no)

**Abstract.** Raindrops interact with water vapour in ambient air while sedimenting from the cloud base to the ground. They constantly exchange water molecules with the environment and, in sub-saturated air, they evaporate partially or entirely. The latter of these below-cloud processes is important for predicting the resulting surface rainfall amount and it influences the boundary layer profiles of temperature and moisture through evaporative latent cooling and humidity changes. However, despite

its importance, it is very difficult to quantify this process from observations. Stable water isotopes provide such information, , as they are influenced by both rain evaporation and equilibration (i.e., the exchange of isotopes between raindrops and ambient air). This study elucidates this option by introducing a novel interpretative framework for stable water isotope measurements performed simultaneously at high temporal resolution in both near-surface vapour and rain. We refer to this viewing device as the $\Delta\delta\Delta d$-diagram, which shows the isotopic composition ($\delta^2\mathrm{H}$, $d$-excess) of equilibrium vapour from precipitation samples

relative to the ambient vapour. It is shown that this diagram facilitates the diagnosis of below-cloud processes and their effects on the isotopic composition of vapour and rain since equilibration and evaporation lead to different pathways in the two-dimensional phase space of the $\Delta\delta\Delta d$-diagram, as investigated with a series of sensitivity experiments with an idealized below-cloud interaction model. The analysis of isotope measurements for a specific cold front in Central Europe shows that below-cloud processes lead to distinct and temporally variable imprints on the isotope signal in surface rain. The influence of

evaporation on this signal is particularly strong during periods with a weak precipitation rate. After the frontal passage, the near-surface atmospheric layer is characterised by higher relative humidity, which leads to weaker below-cloud evaporation. Additionally, a lower melting layer after the frontal passage reduces time for exchange between vapour and rain and leads to weaker equilibration. Measurements from four cold frontal events reveal a surprisingly similar slope of $\frac{\Delta d}{\Delta\delta} = -0.30$ in the phase space, indicating a potentially characteristic signature of below-cloud processes for this type of rain events.

## 1   Introduction

Processes acting during the short travel of rain through the atmosphere from the cloud base to the surface have a maybe surprisingly large relevance for several atmospheric phenomena. The two-phase system of rain and vapour is in constant molecular exchange. In addition, in unsaturated conditions, rain partially evaporates, leading to latent cooling of the air, and moistening

of the boundary layer. Surface rainfall totals may be substantially reduced in cases of strong evaporation (Aemisegger et al., 2015), and in the case of convection in the Sahel, large evaporation-driven cold pools can trigger extensive dust storms known as haboobs (Roberts and Knippertz, 2012). In mid-latitudes, cold pool formation influences low-level moisture convergence and thereby the progression and organisation of convective systems (Bennett et al., 2007).

Measurements of these so-called below-cloud processes (Aemisegger et al., 2015) are challenging. Radiosonde profiles provide instantaneous snapshots of a vertical profile of humidity and temperature, but do not capture precipitation rates, and are expensive when deployed at high frequency. Precipitation radar can continuously provide vertically resolved spectra of rain drops, but does not provide information about relative humidity and temperature, which are necessary to reasonably quantify precipitation evaporation (Xie et al., 2016). Other remote-sensing systems, such as Raman water vapour lidar (Cooney, 1970),
Fourier transform infrared radiometers (Schneider and Hase, 2009) and passive microwave radiometers (Solheim et al., 1998) provide vertical profiles of humidity, but are strongly attenuated during precipitation.

    As a consequence of the lack of sufficient observations, model parameters that represent the interaction of falling raindrops with the air column below the cloud base are poorly constrained. Errors in the representation of this process diminish the model forecast quality due to its impact on the rainfall amount and the dynamics of weather systems. This issue becomes
even more relevant as common weather prediction models, such as COSMO (Steppeler et al., 2003), WRF (Skamarock et al., 2008), or AROME (Seity et al., 2010) progress to resolution beyond the grey zone. At horizontal resolutions below about 10 km precipitation is commonly implemented as a prognostic variable, and convective updrafts, downdrafts and the formation of cold pools are partly resolved at the grid scale. These modelling challenges provide an additional motivation to better understand below-cloud processes.

In this context, stable isotopes of water vapour and rain are useful to investigate below-cloud processes. Stable water isotopes are natural, passive tracers that reflect the phase-change history of water. The stable isotope composition is quantified using isotope ratios, defined as the concentration of the rare (heavy $^2\mathrm{H}^1\mathrm{H}^{16}\mathrm{O}$ or $^1\mathrm{H}_2^{18}\mathrm{O}$) over the abundant (light $^1\mathrm{H}_2^{16}\mathrm{O}$) isotope, e.g.:

$$^2R = \frac{[^2\mathrm{H}^1\mathrm{H}^{16}\mathrm{O}]}{[^1\mathrm{H}_2^{16}\mathrm{O}]} \ . \tag{1}$$

Most studies use the more intuitive $\delta$ notation (Dansgaard, 1964; Galewsky et al., 2016), which expresses the heavy isotope composition of a reservoir in terms of relative deviation of $R$ from an internationally accepted standard:

$$\delta = \frac{R_{\mathrm{sample}} - R_{\mathrm{standard}}}{R_{\mathrm{standard}}} \cdot 1000\%o \ . \tag{2}$$

    A $\delta$ value is defined for both heavy over light isotope concentrations ($\delta^2\mathrm{H}$ and $\delta^{18}\mathrm{O}$) and generally indicated in per mil (‰) relative to Vienna Standard Mean Ocean Water (VSMOW2; IAEA, 2009). As heavy isotopes preferentially condense
due to their larger mass, air subject to rainout subsequently loses heavy isotopes. The increasing depletion with increasing rainout along the trajectory of an air parcel can be approximated by the Rayleigh distillation model (Dansgaard, 1954; Ciais and Jouzel, 1994). Air at higher altitudes and latitudes has on average experienced more cooling and rainout and is thus increasingly depleted of heavy isotopes, reflected in more negative $\delta$-values. As precipitation forms from this vapour depleted

in heavy isotopes, temperature-dependent fractionation will lead to a relative enrichment of heavy isotopes in the hydrometeors. Typically, though, precipitation $\delta$ values will still be depleted relative to the standard ocean water VSMOV2, as expressed in negative $\delta$ values. As rain drops fall through the air column, they continuously exchange water molecules with the surrounding vapour. This exchange is particularly relevant if the air column is at or near saturation. Thermodynamics will direct this exchange towards isotopic equilibrium according to ambient temperature. This process is termed *equilibration* and only occurs when the precipitation is liquid.

In unsaturated conditions, a net transfer of water molecules from the drops to the surrounding air occurs. In addition to the equilibrium fractionation that happens during this transfer, the slower diffusion of the heavy molecules $^2\text{H}^1\text{H}^{16}\text{O}$ and $^1\text{H}_2^{18}\text{O}$ causes non-equilibrium or kinetic fractionation. Thereby, lower relative humidity leads to more intense non-equilibrium fractionation. The second-order parameter $d$-excess ($d = \delta^2\text{H} - 8 \cdot \delta^{18}\text{O}$) is sensitive to such non-equilibrium conditions, since $^2\text{H}^1\text{H}^{16}\text{O}$ reaches isotopic equilibrium faster than $^1\text{H}_2^{18}\text{O}$ (Dansgaard, 1964). The d-excess quantifies the difference in $^2\text{H}^1\text{H}^{16}\text{O}$ and $^1\text{H}_2^{18}\text{O}$ from their ratio expected during equilibrium conditions as a measure of non-equilibrium (Stewart, 1975). Evaporation of rain in unsaturated conditions causes a decrease of d-excess in rain and consequently an increase of d-excess in the surrounding air. Further parameters critically influence this process, such as the drop size distribution (Managave et al., 2016), below-cloud relative humidity (Lee and Fung, 2008), the height of the melting layer, the height of the cloud base (Wang et al., 2016), and vertical wind velocity. Thus, isotopes reflect the conditions that rain drops experience below the cloud, but in a convoluted way that often renders interpretation cumbersome. If stable isotope are to be used for constraining below-cloud processes, different factors need to be disentangled.

Previous studies often investigated only the condensed part of the two-phase system (e.g., Miyake et al., 1968; Celle-Jeanton et al., 2004; Barras and Simmonds, 2009; Risi et al., 2010a; Muller et al., 2015; Managave et al., 2016). These studies sampled rain in high temporal resolution and gave sometimes contrasting explanations for the observed short-term isotopic variations. For example, Coplen et al. (2008) and Yoshimura et al. (2010) investigated an atmospheric river event in California and disagreed on whether below-cloud processes or changes in the formation height caused the variations they observed. Since vapour and rain are in continuous exchange, measuring one without the other makes meaningful interpretation difficult. This is especially the case in situations dominated by advection, for example cold-frontal rain. There, the isotopic evolution of rain is a combined signal of a changing air mass and in-cloud processes, below-cloud equilibration with progressively depleted vapour as the front progresses, and rain evaporation (Dütsch et al., 2016). Simultaneous observations of vapour and precipitation are necessary to distinguish these processes and quantify below-cloud processes. Aemisegger et al. (2015) showed for a mid-latitude rain event that combined observations of stable isotopes in vapour and rain more clearly reveal the influence of below-cloud processes and the structure of the precipitation system.

Thus, joint observations of the stable isotope composition of vapour and precipitation at ground level are valuable recorders of the convoluted influence of several factors and processes. However, extracting the contribution of individual factors is challenging. Here we propose a new set of measures to quantify the influences of equilibration and evaporation on the isotope composition of near-surface vapour and rainfall. To this end, a new interpretative framework is introduced, which allows us to determine the leading below-cloud processes during a precipitation event. This framework is used here to interpret both

high-resolution isotope measurements from cold fronts in Central Europe and results from idealized simulations with a below-cloud interaction model. Section 2 provides information about the measurements and the below-cloud model. The stable water isotope measurements during a cold frontal passage are presented in Section 3. Section 4 introduces the new interpretative framework with an idealized model, before the observations are discussed in the new interpretative framework in Section 5.
Finally, we provide our main conclusions in Section 6.

## 2 Data and Methods

### 2.1 Isotope measurements

Stable water isotopes in ambient water vapour were measured on a tower building at the Institute for Atmospheric and Climate Science of ETH Zurich (47.38°N, 8.55°E; 510 m a.s.l) between 9 October and 27 November 2015 with a cavity ring down
spectrometer (L1115-i, Picarro Inc, USA). Ambient air was directed to the analyser through a 10 m PFA tubing heated to 70°C that was flushed by a bypass pump (HN022AN.18, KNF Neuberger, Germany) with a flow rate of 9 $l$ min$^{-1}$ (Aemisegger et al., 2012; Aemisegger, 2013). The isotopic analyser was calibrated twice a day at ambient humidity levels using a commercial setup (Standards Delivery Module A0101 and Vaporizer V1102-i, Picarro Inc. USA). Two laboratory standards bracketing the composition of typical ambient values in ambient vapour (Standard 1: $\delta^2$H $= -75$‰, $\delta^{18}$O $= -10$‰; Standard 2: $\delta^2$H $=$
$-247$‰, $\delta^{18}$O $= -43$‰) were provided to the analyser for 10 min each. The first 5 min and last 30 s of the calibration, as well as the 10 min ambient air measurements after each calibration were discarded to avoid the influence of memory effect on calibration and the final isotope data. Raw measurements were corrected with an average calibration function from all calibration runs of the measurement period. Frequent gaps in the calibration make this time-independent calibration function more robust compared to the usual linear interpolation between subsequent calibration runs. The thereby neglected shorter-
term drift leads to an increased uncertainty of the calibrated measurements. The 5 s measurements of the instrument were transformed to 10 min average values, which have an average uncertainty after calibration of 1.23‰ for $\delta^2$H, 0.42‰ for $\delta^{18}$O, and 3.6‰ for $d$-excess. For more details about the vapour isotope measurements see Graf (2017).

At the same location, rain was manually sampled during selected events with a simple rainfall collector. The collector consists of a PTFE funnel of 15 cm diameter, which points into a 20 ml glass vial. Each sample was collected in a separate
vial, which was immediately closed after retrieval from the sampler to avoid evaporation after sampling. A default sampling interval of 10 min was applied, which was shortened to 5 min during intense rain, or prolonged up to 30 min if the sampled amount was not sufficient for analysis. The approximate sample amount was recorded, but not used to determine rain rates. The samples were analysed for their isotopic composition in the laboratory with a cavity ring down spectrometer (L2130-i, Picarro Inc., USA) operating for liquid sample analysis (Graf, 2017). The average uncertainty of the calibrated liquid samples
is 1.25‰ for $\delta^2$H, 0.24‰ for $\delta^{18}$O and 1.43‰ for $d$-excess. In this study, 86 continuous rainfall samples collected during a cold frontal passage on 20 November 2015 are presented.

Also measured at the same location were temperature, humidity, wind speed and direction, and precipitation amount and intensity. These parameters were obtained at a 10 min interval from different meteorological sensors (Thygan VTP37 and

wind gauge WN37, meteolabor AG; Tipping bucket rain gauge 7051.1000, Theodor Friedrichs & Co.) on the rooftop with measurement distance of less than 5 m to the ambient air inlet of the isotopic analyser.

## 2.2 Equilibrium vapour from precipitation

Falling rain and the vapour in the atmospheric column below cloud base compose a two-phase system. The constant exchange of water molecules makes the system evolve towards an equilibrium in the isotopic composition of both phases. In isotopic equilibrium, there is no net exchange of isotopologues between the phases. Temperature-dependent isotopic fractionation between light and heavy isotopes however creates different isotopic compositions of the liquid and vapour phases in equilibrium:

$$R_l = \alpha_{v \to l} R_v,\tag{3}$$

which can be equivalently expressed in $\delta$-notation as

$$\frac{\delta_l}{1000} + 1 = \alpha_{v \to l} \left( \frac{\delta_v}{1000} + 1 \right).\tag{4}$$

Here, subscripts $l$ and $v$ denote the liquid and vapour phase, respectively, and $\alpha_{v \to l}$ is the temperature-dependent fractionation factor of the vapour to liquid phase transition. At $20°C$, $\alpha_{v \to l}$ is 1.0850 for $^2H^1H^{16}O/^1H_2^{16}O$ and 1.0098 for $^1H_2^{18}O/^1H_2^{16}O$ (Majoube, 1971b).

We denote the difference due to fractionation between two phases in equilibrium as *equilibrium difference* $\Delta_{l-v}=\delta_l - \delta_v$. Consider, for example, the equilibrium difference for a vapour-liquid system, where the liquid has a composition of $\delta_l = 0‰$ for both $\delta^2H$ and $\delta^{18}O$ (A in Table 1). $\Delta_{l-v}$ for $\delta^2H$ is 78.4‰ at $20°C$ and 101.0‰ at $0°C$. Thus, equilibrium fractionation for cold temperatures is stronger and leads to a larger equilibrium difference of $\delta^2H$ and $\delta^{18}O$. In addition, these differences are smaller if the liquid is more depleted in heavy isotopes. For a liquid with $\delta^2H = -120‰$, $\Delta_{l-v}$ becomes 69.0‰ at $20°C$, and 88.9‰ at $0°C$ (B in Table 1). The increase in fractionation strength with decreasing temperature is stronger for $\delta^2H$ than for $\delta^{18}O$, which leads to a more positive equilibrium difference for $d$ towards colder temperatures. In addition, $d$ of vapour increases and hence the equilibrium difference decreases if the liquid or solid is depleted in heavy isotopes. The dependence of the equilibrium difference on temperature and isotopic composition further complicates matters, in particular for the interpretation of the $d$-excess (Dütsch et al., 2016).

The problem that the comparison of $\delta$-values in precipitation and ambient vapour is not straightforward can be overcome by comparing the isotopic composition of ambient vapour with the *equilibrium vapour from precipitation* for $\delta$-values and $d$, termed $\delta_{p,eq}$ and $d_{p,eq}$ (Aemisegger et al., 2015). The equilibrium vapour from precipitation is calculated as the isotopic composition of vapour that is in equilibrium with rain at ambient air temperature. The direction of isotopic exchange then becomes apparent directly from the difference between $\delta_{p,eq}$ and $\delta_v$ for the $\delta$-values, and from comparing $d_{p,eq}$ and $d_v$ for the $d$-excess. This substantially simplifies the interpretation of the state of equilibrium in the liquid-vapour system. In principle, it would also be possible to introduce in an analogous way an equilibrium precipitation from vapour. We regard the concept of equilibrium vapour as more intuitive below cloud base, and use it here.

In the following, we make use of the isotopic composition of the *equilibrium vapour from precipitation* and denote differences between ambient vapour at the surface and precipitation at any level in the column as:

$$\Delta\delta = \delta^2\mathrm{H}_{p,eq} - \delta^2\mathrm{H}_{v,sfc} \quad \text{and} \tag{5}$$

$$\Delta d = d_{p,eq} - d_{v,sfc}. \tag{6}$$

A $\Delta\delta$ could also be defined for $\delta^{18}\mathrm{O}$, which would require an additional index for $\Delta\delta$ to discriminate between $\delta^2\mathrm{H}$ and $\delta^{18}\mathrm{O}$. Since information about $\delta^{18}\mathrm{O}$ is already included in $d$ the notation is confined to $\Delta\delta$ for $\delta^2\mathrm{H}$. Note that a value of $\Delta d = 0$ does not indicate the absence of non-equilibrium fractionation. It rather is an indication that the ambient vapour and the equilibrium vapour of the precipitation have experienced a similar degree of kinetic effects.

      In the analysis below, we will use $\Delta\delta$ and $\Delta d$ as measures of the deviation of the vapour-precipitation system from equi-
librium. For instance, a negative value of $\Delta\delta$ indicates that precipitation is more depleted in $\delta^2\mathrm{H}$ than ambient vapour, based on the equilibrium difference at ambient temperature discussed above. It will be shown that this results in a powerful, intuitive interpretative framework (referred to as the $\Delta\delta\Delta d$-diagram) to quantify physical processes between the cloud base and the surface from highly-resolved stable isotope measurements in water vapour and precipitation. The interpretation of this new diagram will be further substantited with results from idealized simulations with a below-cloud interaction model, introduced
in the next subsection.

## 2.3    Below-cloud interaction model

In order to support the interpretation of isotope measurements with our new framework and to quantify the role of different processes, we apply a one-dimensional below-cloud interaction model. The model simulates the microphysical and isotopic interactions of a falling hydrometeor with the surrounding air, as described in detail in Appendix A and Graf (2017). In this
section, we lay out its general setup and initialisation.

      The model consists of a single vertical column that extends from the ground to the height where a single hydrometeor is introduced. The hydrometeor falls through the column with its terminal velocity, grows or evaporates, changes its temperature and isotopically equilibrates with the surrounding vapour. Isotope processes are parameterized following Stewart (1975) with separate mass balance equations for all three isotope species (Appendix A4). Interactions with other hydrometeors (collision
and breakup) are neglected. Horizontal and vertical air motion are also neglected, i.e., there is no horizontal advection into or out of the column, and no up- or downdraft. As output, the model yields vertical profiles of the hydrometeor size and its isotopic composition.

      Profiles of temperature, humidity and the isotopic composition of the surrounding vapour have to be provided to the model as input prior to the initialisation with rain. These initial profiles can be specified in two ways: (i) based on measurements or sim-
ulations with isotope-enabled atmospheric models (such as COSMOiso, Pfahl et al., 2012), or (ii) calculated from the idealised (moist) adiabatic ascent of an air parcel from the surface to the top of the model column with a Rayleigh fractionation process after reaching saturation (Appendix A). The profiles are assumed to be unaffected by the falling hydrometeor throughout the

simulation. This assumption only holds if a single hydrometeor is considered. When simulating rain events during which many hydrometeors fall and subsequently affect the surrounding air, the assumption becomes invalid over time. A remedy to this problem would be to reinitialize the model regularly with updated profiles of the surrounding air.

The hydrometeor size is defined as the *equivalent liquid diameter*, which corresponds to the diameter of a spherical liquid drop with the same mass as the hydrometeor. The model can be initialised with a pre-defined hydrometeor size at the height of initialisation. Alternatively, as used in this study, the terminal diameter at the surface can be provided as input. In this case, the hydrometeor size at the height of initialisation is varied iteratively until the target diameter at the surface is reached. To simulate bulk precipitation, the model can be run (i) for all bins of a drop size distribution, which are then used to calculate a mass and number-weighted sum or (ii) for just one hydrometeor size, which approximates the drop size distribution with a single diameter. This is represented by the mass weighted mean diameter $D_m$, which is obtained in this study from the rain rate by assuming a Marshall-Palmer distribution.

The initial isotopic composition of the hydrometeor is determined by the surrounding vapour at its initialisation height. By default, formation via the Wegener-Bergeron-Findeisen mechanism is assumed between 0 and $-23°$C. Optionally, a fraction of mass can be added that is formed by riming of supercooled cloud droplets on the hydrometeor (Appendix A2). The hydrometeor is solid at temperatures below 0°C and melts instantaneously when its temperature exceeds 0°C. Although melting happens over a $\sim$300 m deep layer in reality (Frick et al., 2013), this is a valid assumption considering that hydrometeors start to melt from the outside (Austin and Bemis, 1950) and therefore expose their liquid fraction to the surrounding vapour from the beginning of the melting process.

## 3   Cold frontal passage on 20 November 2015

We now apply the framework outlined above to data from a prolonged rainfall period in northern Switzerland. High-resolution rain and vapour isotope measurements reveal variations in the below-cloud processes during the event.

### 3.1   Meteorological situation

The local meteorology of this event was characterised by an extended front over Central Europe, which was the remnant of a cold front associated with a decaying cyclone over the Gulf of Finland. The nearly zonally oriented front passed Switzerland from a northerly direction during 20 November 2015, before leading to the genesis of a new cyclone over the Gulf of Genoa on the following day. The rainband associated with the cold front extended zonally over a distance of about 400 km from the Burgundy (France) across Switzerland to the Lake Constance, with a distinct band of high rain intensity (Fig. 1a). This intense rainband was embedded in a broader zone with stratiform rain. Near Zurich, the frontal passage led to a decrease of equivalent potential temperature ($\theta_e$) at 850 hPa of more than 12 K and to a veering of the wind from southwest to northwest (Fig. 1b).

## 3.2 Meteorological surface observations

An overview of selected surface measurements between 06 UTC 20 November and 01 UTC 21 November 2015 is shown in Fig. 2. The local 2-m temperature ($T$; red line in Fig. 2a) remained roughly constant during the first part of the event, with a slight increase before 14 UTC. At 19 UTC, when the surface front arrived at the measurement location, a rapid drop of about $2.5°C$ in 30 min was recorded. The temperature gradually decreased further by about $3.5°C$ between 20 and 22 UTC and remained constant thereafter, resulting in an overall decrease in $T$ of $\sim 6$ K. Local relative humidity at 2 m ($h$; blue line in Fig. 2a) varied between $75 - 85\%$ before the frontal passage, and increased to values around $85 - 90\%$ thereafter.

The rain associated with this frontal event started in Zurich at 06 UTC 20 November and lasted until 03 UTC 21 November 2015. Most of the precipitation appeared to be of stratiform nature. The total rain measured by a rain gauge on the rooftop was 30.9 mm, whereof 27.5 mm fell during the part of the event investigated here (07:00 - 23:30 UTC). The intensity varied between 0 and $3\,\mathrm{mm\,h^{-1}}$, before increasing briefly to $10\,\mathrm{mm\,h^{-1}}$ as the surface front passed at 19 UTC (Fig. 2b). Thereafter, the intensity remained relatively high compared to the period prior to the frontal passage with an average of $3\,\mathrm{mm\,h^{-1}}$ until approximately 23 UTC, when it decreased to low values for the remainder of the event. Between 12 and 18 UTC, sustained wind speeds occurred with values between 5 and $10\,\mathrm{m\,s^{-1}}$, and therefore the rain intensity is likely underestimated during this period due to the exposed location of the rain gauge. A less exposed meteorological station (MeteoSwiss Station Zurich Fluntern, at a distance of 1.3 km) recorded a total amount of rain of 38.3 mm at 1 m above ground level. For further analysis, we split the event into a pre-frontal period until about 18:45 UTC (purple shading, precipitation samples 1–54) and a post-frontal period thereafter (green shading, precipitation samples 55–86).

According to two balloon soundings launched from the measurement site in Zurich during the event, the height of the $0°C$ isotherm decreased from 2700 m a.s.l. at 16:30 UTC to 1500 m a.s.l. at 22:30 UTC during the frontal passage (not shown).

## 3.3 Isotopic composition of vapour and rain

The 10-minute averaged isotope values in surface vapour in Zurich were between $-265‰$ and $-105‰$ for $\delta^2\mathrm{H_v}$ (Fig. 2c, black line), and between $-35‰$ and $-14‰$ for $\delta^{18}\mathrm{O_v}$ (not shown). The vapour isotope measurements exhibit an overall decrease of more than $160‰$ for $\delta^2\mathrm{H_v}$ during the entire event. A weak decrease in $\delta^2\mathrm{H_v}$ around 08 UTC was followed by a steady increase until 14 UTC. $\delta^2\mathrm{H_v}$ decreased thereafter, and the decrease became steeper after 18 UTC, before reaching a roughly constant minimum value at 23 UTC of about $-265‰$. For $d_\mathrm{v}$, values increased from $5‰$ to $20‰$ during the event (Fig. 2d, black line). A gradual increase by about $5‰$ before the arrival of the surface front was followed by a more rapid $10‰$ increase in the 4 h after the frontal passage at about 19 UTC, marked by a distinct spike of $5‰$ in $d_\mathrm{v}$. Other short-term variations of $d_\mathrm{v}$ were within the uncertainty range (grey shading).

To identify the possible influence of below-cloud processes we now compare the vapour isotope measurements with the precipitation, using the above-defined metric of equilibrium vapour. The isotopic signals of vapour ($\delta^2\mathrm{H_v}$; black line in Fig. 2c) and equilibrium vapour from the 86 rain samples ($\delta^2\mathrm{H_{p,eq}}$; blue bars in Fig. 2c) exhibit a similar evolution during the whole event. Differences are overall less than $23‰$. $\delta^2\mathrm{H_{p,eq}}$ is more variable and its evolution is less smooth than for $\delta^2\mathrm{H_v}$. After

an initial decrease with a subsequent increase similar to $\delta^2 H_v$, $\delta^2 H_{p,eq}$ reaches two maxima at around 14 and 16 UTC, which coincide with low relative humidity and weak rain intensity. It decreases afterwards until the end of the sampling period. The decrease is particularly strong during the passage of the surface front and during the second distinct temperature drop (after 20:30 UTC). The overall evolution corresponds to a flat W-shape in the first part of the event until 16 UTC, and a strong decrease in the second part. This is similar to what Dütsch et al. (2016) found for a cold front in an idealised extratropical cyclone, but in our case without the increasing branch at the end, which may have occurred during weak rain at the end of the event (not sampled).

The $d_{p,eq}$ varies around 0‰ before 19 UTC, and then increases markedly during the passage of the front with values of more than 10‰ (Fig. 2d). Notably, negative values of $d_{p,eq}$ occur during periods with weak rain (e.g., around 08:30, 13:30 and 16:00 UTC). $d_v$ also increases during the event, but less abruptly and with less variations than for $d_{p,eq}$, which exhibits a positive correlation with $h$ (Spearman $\rho = 0.88$) and rain intensity ($\rho = 0.63$). Smaller drops during phases with weak rain and low relative humidity experience enhanced evaporation, which decreases $d_{p,eq}$.

The similar evolution of $\delta^2 H_v$ and $\delta^2 H_{p,eq}$ in Fig. 2c indicates that equilibration of rain with the surrounding vapour plays an important role for the evolution of the time series. Alternatively, part of the vapor sampled at the surface could have been transported downwards from cloud formation levels by convective downdrafts. In the case analysed here, this influence may be limited due to the mainly stratiform character of the event. Nonetheless, it remains a principal challenge to identifying the influence of below-cloud processes in joint observations of vapour and precipitation. One example are signals from meso-scale meteorological processes, such as the transition between airmasses at the weather front. In order to facilitate the interpretation of these measurements in terms of below-cloud processes, we introduce in the next sections a new framework that makes the involved physical processes more explicit.

One can also consider the effect of below-cloud processes on ambient vapour. However, on short enough time-scales (a hydrometeor falling from the cloud to the ground), the effect on vapour can be neglected, since the amount of vapour in a given air volume exceeds the amount of liquid or solid by far, especially for the rain rates we measured (for the calculation see Graf (2017)). The effect on vapour would only appear over a longer time period. In the event we present here, a part of the gradual depletion of vapour after 16 UTC could be caused by interaction with falling precipitation or downward motion of the air, which introduces depleted moisture.

It is apparent from Fig. 2c,d that the difference between vapour isotope measurements and the equilibrium vapour for precipitation varies systematically throughout the precipitation event. Their difference is conveniently quantified by $\Delta\delta$ for $\delta^2 H$ (Eq. 5), and correspondingly by $\Delta d$ for $\delta d$ (Eq. 6). The time series of $\Delta\delta$ for all precipitiation samples from the frontal event varies between -20 permil and 12‰ (Fig. 3a). For $\Delta d$, the time series shows negative values, except for the passage of the front (Fig. 3b). Some rain samples are in equilibrium with vapour for $\delta^2 H$ ($\Delta\delta \simeq 0$‰; e.g., at 15 UTC), for $d$ ($\Delta d \simeq 0$‰; at about 19 and 21 UTC) or for both ($\Delta\delta$ and $\Delta d \simeq 0$‰; at 10 UTC). Other samples indicate the influence of below-cloud evaporation with a positive $\Delta\delta$ and a strongly negative $\Delta d$ (at about 14 and 16 UTC). Most post-frontal samples have a strongly negative $\Delta\delta$ and a $\Delta d$ close to zero, which indicates the conservation of depleted $\delta^2 H_{p,eq}$ from the cloud and incomplete equilibration

with near-surface vapour. The influence of rain evaporation also results in a negative correlation of $\Delta\delta$ with $h$ ($\rho = -0.65$) and rain intensity ($\rho = -0.44$). The correlation with $h$ is also strong for $\Delta d$ ($\rho = 0.83$).

## 4 Idealized simulations with a below-cloud interaction model

The systematic variation of $\Delta\delta$ and $\Delta d$ throughout the precipitation event motivates us to investigate the influence of me-
teorological driving factors on these parameters using an idealised model of below-cloud effects (Sec. 2.3). To illustrate the representation of below-cloud processes in this model, we in detail consider the isotopic fractionation of falling precipitation in a set of reference simulations and sensitivity experiments, before transferring the findings to the measurements of the precipitation during 20–21 November 2015.

### 4.1 Reference simulations

The model configuration consists here of a single-column model domain with a surface pressure of 950 hPa, and extending from 500 m at the surface to 3500 m a.s.l. Time-constant vertical background profiles of temperature $T$, relative humidity $h$, $\delta^2H_v$ and $d_v$ are obtained from the moist adiabatic ascent of an air parcel that is lifted from the surface with initial values of $T_0 = 12°C$, $h_0 = 0.75$ (Fig. 4a, green and blue lines). The background isotope profiles are obtained correspondingly from Rayleigh fractionation during a moist adiabatic ascent with an surface composition of $\delta^2H_v = -150‰$ and $d_v = 10‰$ (Figs.
4c,d, solid black lines). Below cloud base (lifting condensation level) at 1030 m a.s.l (dotted horizontal lines), specific humidity and isotopic composition of the vapour are constant, while $h$ increases. Above cloud base, the air parcel follows a Rayleigh fractionation process. Fractionation increases with decreasing temperature and hence the rate of decrease of $\delta^2H_v$ becomes more negative with height. The effect of condensation on the profile of $d_v$ (black line in Fig. 4d) is small at low altitudes and only becomes apparent in the uppermost 500 m of the domain, where $d_v$ starts to increase. Note that this background state of
the model is not affected by evaporating droplets or other processes during the simulation.

Now, three hydrometeors representing typical drop sizes for mid-latitude rain are introduced at the formation height at 3500 m a.s.l. The initial diameters (0.56, 1.02 and 2.00 mm) have been calculated iteratively such that the hydrometeors reach target diameters of 0.5, 1 and 2 mm when arriving at the surface. The hydrometeors fall with an average terminal velocity of 2.4, 4.2 and 7.0 m s$^{-1}$, respectively, while growing in supersaturated and shrinking in unsaturated conditions, as expressed by
their mass relative to the mass at formation height $m/m_{init}$ (Fig. 4b). The saturation of the environment with respect to the hydrometeor depends on the phase and the temperature of the hydrometeor, quantified by the effective relative humidity $h_{eff}$ of a 1 mm hydrometeor (Fig. 4a, dotted blue line). The air layer between formation height (3500 m) and the 0°C-isotherm ($\sim 2250$ m) is saturated with respect to liquid water and supersaturated with respect to ice. Therefore, solid hydrometeors grow due to $h_{eff} > 100\%$. The growth slows down as $h_{eff}$ becomes smaller towards the 0°C-isotherm, but continues between
the 0°C-isotherm and the cloud base as hydrometeors fall into warmer air and retain a slightly lower temperature than the environment. Finally, the hydrometeors fall into sub-saturated air below the cloud base and start to evaporate. The decreases of $m/m_{init}$ is fastest for the small hydrometeor (Fig. 4b, blue line). Evaporation decreases the droplet temperature, which leads

to a higher $h_{\mathrm{eff}}$ than $h$ below the cloud base. This effect dampens evaporation by more than 50% compared to a case where the droplet takes on ambient air temperatures.

The initial isotopic composition of the hydrometeors (Fig. 4c, solid colored lines, symbol A) is enriched by about 100‰ in $\delta^2\mathrm{H}$ compared to the composition of the surrounding vapour (black line). Above the 0°C-isotherm, the hydrometeors are frozen and thus hardly change their isotopic composition (Figs. 4c,d; A to B). Simulated hydrometeors melt instantaneously when their temperature exceeds 0°C and equilibration sets in, which rapidly changes their isotopic composition towards equilibrium with the surrounding vapour. Comparison between the isotopic composition of the droplets (Figs. 4c,d, solid coloured lines, symbols A,B,C) and the background vapour is facilitated here by using the equilibrium variables $\delta_{\mathrm{p,eq}}$ and $d_{\mathrm{p,eq}}$ (dashed coloured lines, symbols A',B',C'). A drawback of these variables is the discontinuity at the height of the 0°C-isotherm (Figs. 4c,d, symbol B'). When the hydrometeor changes its state from solid to liquid, the fractionation coefficients change and consequently $\delta_{\mathrm{p,eq}}$ and $d_{\mathrm{p,eq}}$ jump.

Hydrometeors equilibrate more quickly the smaller they are, while the 2 mm hydrometeor never reaches equilibrium. Below cloud base, evaporation leads to an enrichment of the small hydrometeors with respect to equilibrium with the surrounding vapour (symbol C' in 4c). The hydrometeors' $d$-excess is smaller than $d_{\mathrm{v}}$ (Fig. 4d, solid lines at symbol A, black line). Non-equilibrium fractionation due to supersaturation with respect to ice increases $d_{\mathrm{p}}$ compared to $d_{\mathrm{v}}$ (symbol C in Fig. 4d). The smaller $d_{\mathrm{p}}$ values found here are due to the fact that for strongly depleted vapour, the equilibrium fractionation of $\delta^2\mathrm{H}$ is less than 8 times stronger than that of $\delta^{18}\mathrm{O}$, as discussed in detail by Dütsch et al. (2017).

## 4.2 Reference simulations in the $\Delta\delta$ and $\Delta d$ diagram

We will now cast the results from the idealised model using the variables $\Delta\delta$ and $\Delta d$ that have been introduced above to measure the deviation of the precipitation from equilibrium with ambient vapour at the surface. To this end, we consider first the $\Delta\delta$ in the reference simulations above for three different rain drops that fall through the atmospheric column (Fig. 5a). After formation at a height of $z = 3500$ km (colored diamonds), the hydrometeors are depleted by 63‰ in $\delta^2\mathrm{H}$ (i.e., $\Delta\delta$ is -63‰) compared to surface vapour. This is both a contribution from the depletion of the background vapour profile (-75‰). For simplicity, we only show vapor above liquid, which results in a $\Delta\delta$ of -63‰. As the droplets fall, the $\Delta\delta$ changes little until reaching the melting level (colored triangles). Equilibration above cloud base (coloured crosses) moves them progressively closer to the ambient vapour (black line) and its surface value (black circle). Below cloud base, evaporation in addition introduces fractionation that leads to positive $\Delta\delta$ for the smallest droplet (blue line), whereas the largest droplet has a negative $\Delta\delta$ at the surface, indicating incomplete equilibration, that was not overprinted entirely by the evaporation-induced fractionation.

The initial $\Delta d$ of -7.5‰ at formation height evolves due to both equilibrium and kinetic fractionation as the droplets fall through the atmospheric column (Fig. 5b). This leads initially to $\Delta d$ becoming less negative, reaching equilibrium with the ambient vapour for the smallest droplets at cloud base. As the droplets continue to fall through an unsaturated atmosphere below, kinetic fractionation sharply increases $\Delta d$, again most markedly for the small droplets, which experience the strongest relative loss of their mass.

When using $\Delta\delta$ and $\Delta d$ as the axes of a new diagram, the evolution of the droplets in the three reference simulations yield inverted U-shaped curves (Fig. 5c). In the examples provided here, these curves depend entirely on the size of the rain drops at the surface (large filled dots), placing them either in the lower left quadrant of the diagram (large drop, comparatively weak below-cloud interaction) or in the lower right quadrant (small drop, with at first complete equilibration followed by strong below-cloud evaporation). Hence, the location of a precipitation sample in this $\Delta\delta\Delta d$-diagram is determined by several processes that occur along the trajectories of the rain drops from their formation until they are measured at the surface. The origin of the diagram ($\Delta\delta = 0\permil, \Delta d = 0\permil$) indicates full equilibrium between vapour and precipitation. Note that this does not indicate that the involved vapour and rain drops did not experience non-equilibrium fractionation processes; it merely indicates that at the time of simultaneously measuring water isotopes in vapour and rain, the two values correspond to the local equilibrium conditions.

Note that the measured data points of $\Delta\delta$ and $\Delta d$ shown in Fig. 5c can be compared with the values from our idealized simulations at the final (surface) location shown in Fig. 3. Therefore, we now display the measurement data points in the $\Delta\delta\Delta d$-diagram to investigate the influence of different below-cloud processes on the surface measurements during the frontal passage. By means of additional model sensitivity experiments, we then apply this framework to interpret and quantify the influence of below-cloud effects on the vapour and precipitation isotope composition observed at the surface during the frontal passage in November 2015.

## 5   Observed below-cloud effects in the $\Delta\delta\Delta d$-diagram

### 5.1   Rain samples during the cold frontal passage

The 86 rain samples cover a much larger range in the $\Delta\delta\Delta d$-diagram than the 3 idealised simulations (Fig. 6a). Some data points are located in the lower right quadrant, associated with intermediate rain rates (cf. Fig. 2b) during the pre-frontal phase of the event (blue to green shading). Compared to the idealised simulations, these data points match with intermediate to small droplets that experienced evaporation (blue and red dot). Data points located to the left of the origin indicate that precipitation is more depleted than ambient vapour, and reflect that more of the initial signal after formation ("cloud signal") is retained in precipitation. In the idealised experiments, this corresponds to the largest drop size (yellow dot). Most of the post-frontal data points with the most intense rain rates (cf. Fig. 2b) are located to the left of the origin (orange to red shading).

Drop size, and thus rain rate appears as an important driving factor of the below-cloud processes. Figure 6b shows another variant of the $\Delta\delta\Delta d$-diagram where the dot size indicates rain rate. It appears that samples with the highest rain rates are located in the upper left corner, as they are least affected by below-cloud processes and retain more of their initial strongly negative $\Delta\delta$. Samples from periods with weak rain rates are located in the bottom right corner of the diagram, reflecting a stronger evaporation influence. Overall, complete equilibration with ambient vapour seems to be rather limited because only few data points are close to the origin of the diagram. The regions in Fig. 6a that are covered by pre-frontal (purple) and post-frontal (green) samples are fairly well separated. Pre-frontal samples, which are on average higher in $\Delta\delta$ and lower in $\Delta d$, seem to be more strongly affected by below-cloud processes than post-frontal samples. From the idealised model experiments,

such a difference can be explained by an on average lower rain intensity and a lower relative humidity during the pre-frontal phase, and therefore by enhanced below-cloud equilibration and evaporation. Additionally, the melting layer was clearly lower after the passage of the front, and thus both vertical distance and time for equilibration were reduced. Post-frontal samples therefore carry more of their depleted initial $\delta^2\mathrm{H}_{\mathrm{p,eq}}$ from the cloud base to the surface.

The data points in Fig. 6 roughly fall along a line with a negative slope. A linear fit through the samples yields a regression line with a slope of $\frac{\Delta d}{\Delta \delta} = -0.31$ (Fig. 6b, solid black line). It is noteworthy that similar slopes ($-0.30 \pm 0.02$; dashed black lines in Fig. 6b) were found for three other cold fronts in Switzerland (Graf, 2017). This indicates that the slope could represent a general characteristic of below-cloud evaporation and equilibration of rainfall, at least for continental mid-latitude cold front passages. It will be insightful to explore the slope in the $\Delta\delta\Delta d$-diagram for other climatic regions in future studies.

It is important to recall that the isotopic evolution of sedimenting rain drops is strongly influenced by ambient meteorological conditions, in particular the detailed relative humidity profile, the formation height of precipitation, the isotope profile of vapour, and potential up- and downdrafts and turbulent motions below the cloud base. The effect of some of these processes is now investigated with the aid of the idealised below-cloud interaction model, providing further insight to the interpretation of our measurements in the $\Delta\delta\Delta d$-diagram.

**5.2    Sensitivity experiments in the $\Delta\delta\Delta d$-diagram**

We now use the below-cloud interaction model to assess the relevance of different ambient conditions for the rain drop trajectories and surface arrival points in the $\Delta\delta\Delta d$-diagram. Explored parameters include the sensitivity to surface relative humidity, surface temperature, formation height, riming, and the background isotope profiles in terms of $\delta^2\mathrm{H}$ and $d$ (as described in detail in Graf (2017)). For each parameter, several simulations were performed for a range of drop sizes from 0.6 to 1.8 mm.

Assuming a standard Marshall-Palmer dropsize distribution, these diameters correspond to the mass-weighted mean diameter for rain intensities in the range from 0.1 to $20\,\mathrm{mm\,h^{-1}}$.

     For a particular setup of the ambient parameters, the different $\Delta\delta$ and $\Delta d$ when the drops arrive at the surface are connected by dashed lines in Fig. 7. The black line shows the reference experiment (REF, cf. Sec. 4.2), where the filled circle corresponds to the highest rain intensity, and the triangle to the lowest. The label of the experiments points to the input parameter that

is modified. RH50 and RH100 correspond to sensitivity experiments with different surface relative humidity $h_0 = 50\%$ and $h_0 = 100\%$, respectively. T7 and T17 denote experiments with different surface temperatures $T_0 = 7°\mathrm{C}$ and $T_0 = 17°\mathrm{C}$, FH2.5 and FH5.0 refer to experiments with formation heights of $2.5\,\mathrm{km}$ and $5.0\,\mathrm{km}$ a.s.l., respectively, and RIM corresponds to formation by riming. Experiments with altered background profiles of stable water isotopes are denoted as $\delta \pm 50$ ($\delta^2\mathrm{H}_{\mathrm{v}}$ profile changed by $\pm 50\text{‰}$ above the cloud base) and $d \pm 10$ ($d_{\mathrm{v}}$ profile changed by $\pm 10\text{‰}$ above the cloud base).

Changes of the model input parameters systematically affect the position and orientation of the curves in the $\Delta\delta\Delta d$-diagram. The results for simulations where the initial composition of hydrometeors is modified (T7, T17, FH2.5, FH5.0, RIM, $\delta \pm 50$ and $d \pm 10$) diverge for strong rain intensities. For small drops, i.e., weak precipitation intensities, however, the results converge and are quite similar for all simulations. This agrees with the finding from the reference simulations that below-cloud interaction affects samples from weak rain more strongly and overwrites initial differences. Simulations that alter the extent of below-cloud

interaction (RH50, RH100 and to a small degree also T7) show large differences for small drops. For example, evaporation in RH50 shifts isotope values in small drops to high $\Delta\delta$ and low $\Delta d$. In contrast, the absence of evaporation in RH100 leads to an almost complete equilibration with the ambient vapour and almost no change of $d$. Large drops, representative of strong rain intensities, carry a stronger imprint of the different initial composition of precipitation to the surface. Therefore, the coloured dots from simulations with a low initial $\delta^2H$ (T7, FH5.0, $\delta - 50$) are located at lower $\Delta\delta$ than simulations with a high initial $\delta^2H$ (T17, FH2.5, $\delta + 50$). The same is the case for $\Delta d$ in simulations where the initial $d$ differs.

The set of idealized simulations reveals that the closer a precipitation sample is to the origin of the coordinate system, the more it has equilibrated with ambient vapour until it reaches the ground, while remaining unaffected by evaporation. Samples that encountered significant evaporation during their fall are located towards the bottom right of the $\Delta\delta\Delta d$-diagram. This is typically the case for samples from weak rain intensities. Samples that were weakly influenced by equilibration or evaporation during their fall, which is typically the case for intense precipitation, are located towards the left side of the diagram. Assuming constant ambient conditions, variations of the rain intensity cause variations in the $\Delta\delta\Delta d$-diagram along a curve as indicated in Fig. 7. The location and orientation of this curve in this diagram is determined by the meteorological conditions. Studying the evolution of precipitation samples in the $\Delta\delta\Delta d$-diagram during a rain event can thus clearly reveal information about the prevailing meteorological conditions and their temporal evolution.

## 6 Conclusions

The processes acting on precipitation as it falls from the cloud base to the surface are complex and difficult to access from surface measurements only. Using highly resolved measurements of stable isotopes in vapour and rain at the surface, we show here that it is possible to identify an integrated signal of these so-called below-cloud processes when comparing the isotopic composition of equilibrium vapour from precipitation relative to near-surface vapour simultaneously for both $\delta^2H$ and $d$.

We combine this information in a new interpretation framework, the $\Delta\delta\Delta d$-diagram, where $\Delta\delta$ is shown along the x-axis and $\Delta d$ along the y-axis. This combines a view of $\delta^2H_v$, $\delta^2H_{p,eq}$, $d_v$ and $d_{p,eq}$ while tuning down the influence of first-order advection processes during a frontal transition. To display data in the $\Delta\delta\Delta d$-diagram, the isotopic composition of surface vapour and precipitation have to be known, as well as surface temperature.

A $\Delta\delta\Delta d$-diagram shows the isotopic composition of equilibrium vapour from precipitation samples relative to the ambient surface vapour at the time when the samples were taken. By means of idealised below-cloud interaction model simulations, we show that the location of a precipitation sample in the $\Delta\delta\Delta d$-space is determined by two factors: (i) the initial composition of precipitation after formation in the cloud and (ii) the modification of this composition below the cloud by equilibration and evaporation. These below-cloud processes depend on the rain intensity: larger drops during intense rain are typically less affected by below-cloud processes because they spend less time in the air due to a faster fall velocity and they are less affected by exchanges with the ambient vapour due to a smaller surface to volume ratio. The isotopic composition of a rain sample is a mass weighted average of the composition of all drops contained in a sample. The processes that act on a single drop are thus

directly relevant for bulk precipitation. The usefulness of this diagram is demonstrated with measurements from a cold frontal rain event in Switzerland in November 2015.

The main conclusions from this study are:

1. Equilibration between vapour and rain and evaporation of rain in unsaturated air leave distinct imprints on the isotope signal of surface rain. Both aspects of the exchange between the liquid and solid phase become more accessible by quantifying the deviation from isotopic equilibrium with the surface vapour by studying the two quantities $\Delta\delta$ and $\Delta d$.

2. The $\Delta\delta\Delta d$-diagram facilitates the interpretation of the effects of below-cloud processes on rain samples by jointly displaying the degree of equilibration between rain and vapour and the influence of evaporation using the newly defined variables $\Delta\delta$ and $\Delta d$. Equilibration and evaporation have different pathways in the $\Delta\delta\Delta d$-diagram, which makes them more easily distinguishable than in a time series. Investigating rain samples in the $\Delta\delta\Delta d$-diagram can therefore complement a time-series perspective.

3. During the 20 November 2015 cold frontal rainfall event evaporation appears as the dominant below-cloud process regarding the isotopic composition of surface rain. The effect of evaporation on the isotope composition is strongly modulated by the rain rate. The pre-frontal period with weaker rainfall therefore experienced a stronger signal of evaporation below cloud base, whereas the more intense post-frontal rainfall contained a stronger signal from the cloud level. The cloud signal was also more preserved due to higher below-cloud relative humidity, and a lower temperature and melting layer after the frontal passage.

4. In the $\Delta\delta\Delta d$-diagram, below-cloud processes caused precipitation measurements to follow a line with a negative slope of $\frac{\Delta d}{\Delta\delta} = -0.31$. Similar slopes were obtained for several other frontal rain events, suggesting that the characteristics of below-cloud processes, as revealed by the $\Delta\delta\Delta d$-diagram are similar for this type of cold frontal rain events in continental mid-latitudes.

Using the $\Delta\delta\Delta d$ framework, it will be highly valuable to investigate below-cloud effects for other precipitation events. For example, a snowfall event, or a transition from rain to snow could show a stronger cloud signal due to the absent exchange between vapour and solid. Cases of drizzle could exhibit a large degree of equilibration between small drops and ambient vapour. Cases of convective rainfall could show variations due to more cloud-related signals in convective downdrafts.

Further constraints on observations from radiosondes, vertically resolved isotope measurements using aircraft (e.g., Dyroff et al., 2015; Sodemann et al., 2017) and related measurements at high resolution will provide possibilities to validate and apply the idealised modelling framework presented here for below-cloud processes.

We expect that the analysis of the isotopic composition during rain events at other locations and further model studies will benefit from using the parameters $\Delta\delta$ and $\Delta d$, and the $\Delta\delta\Delta d$-diagram as an additional viewing device to obtain insight into below-cloud processes. Thereby, further constraints on microphysical processes in models can be obtained, and ultimately contribute to a more complete use of stable water isotopes to build internally consistent water cycles into numerical weather prediction and climate models.

**Table 1.** Calculated difference in isotopic composition of liquid ($\Delta_{l-v}$) or solid ($\Delta_{s-v}$) in equilibrium with a vapour of different isotopic composition. The fractionation factors of Majoube (1971b) are used for the calculations.

| composition of vapour | | $\Delta_{l-v}$ @ 20°C | $\Delta_{l-v}$ @ 0°C | $\Delta_{s-v}$ @ 0°C |
|---|---|---|---|---|
| $\delta^2$H | −80.0‰ | 78.2‰ | 103.3‰ | 121.3‰ |
| $\delta^{18}$O | −10.0‰ | 9.7‰ | 11.6‰ | 15.1‰ |
| $d$ | 0.0‰ | 0.7‰ | 10.5‰ | 0.6‰ |
| $\delta^2$H | −200.0‰ | 68.0‰ | 89.9‰ | 105.4‰ |
| $\delta^{18}$O | −25.0‰ | 9.5‰ | 11.4‰ | 14.9‰ |
| $d$ | 0.0‰ | −8.4‰ | −1.6‰ | −13.4‰ |

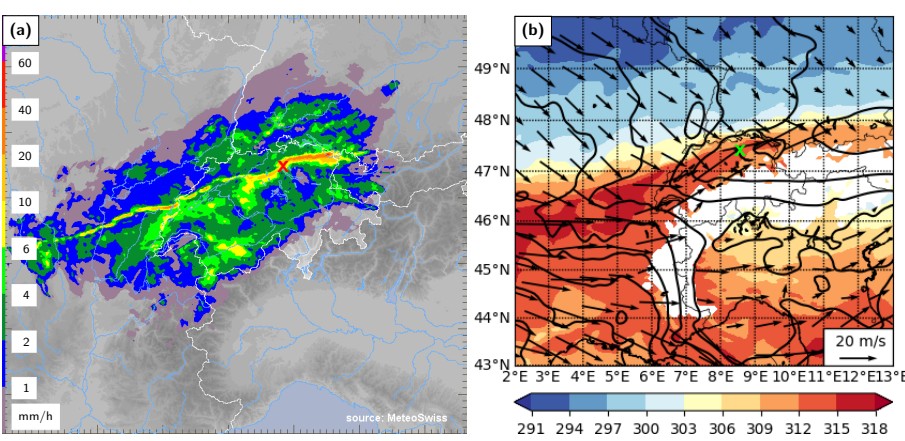

**Figure 1.** (a) Radar composite of surface rain intensity from MeteoSwiss at 19 UTC 20 November 2015, when the surface front passed over the measurement site. (b) Equivalent potential temperature ($\theta_e$ in K, colour) and horizontal wind (arrows) at 850 hPa from COSMO-2 analysis data at 19 UTC 20 November 2015. The location of the measurement site at Zurich is indicated by a red and a green cross, respectively.

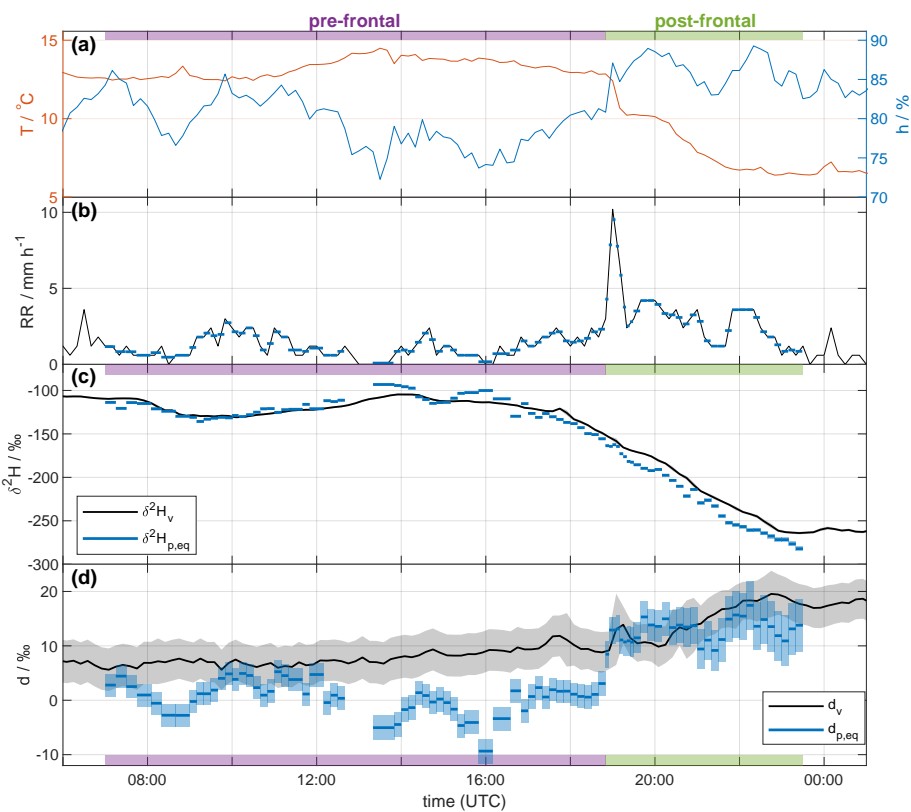

**Figure 2.** Time series of observations in Zurich between 06 UTC 20 November and 01 UTC 21 November 2015. (a) Local temperature ($T$; red) and relative humidity ($h$; blue) measured by the meteorological station. (b) Rain intensity from the rain gauge (black). Blue bars indicate the average values for each rain sample period. (c) $\delta^2\mathrm{H}$ of near-surface vapour (10 min averaged $\delta^2\mathrm{H_v}$; black line) and of the equilibrium vapour from precipitation ($\delta^2\mathrm{H_{p,eq}}$; blue bars). The width of the blue bars denotes the period over which the rain samples were collected. (d) Same as in (c), but for $d$. The calibrated uncertainties are indicated by the shaded areas (hardly or not visible for $\delta^2\mathrm{H_v}$ and $\delta^2\mathrm{H_{p,eq}}$). Pre- and post-frontal periods are indicated with purple and green bars, respectively.

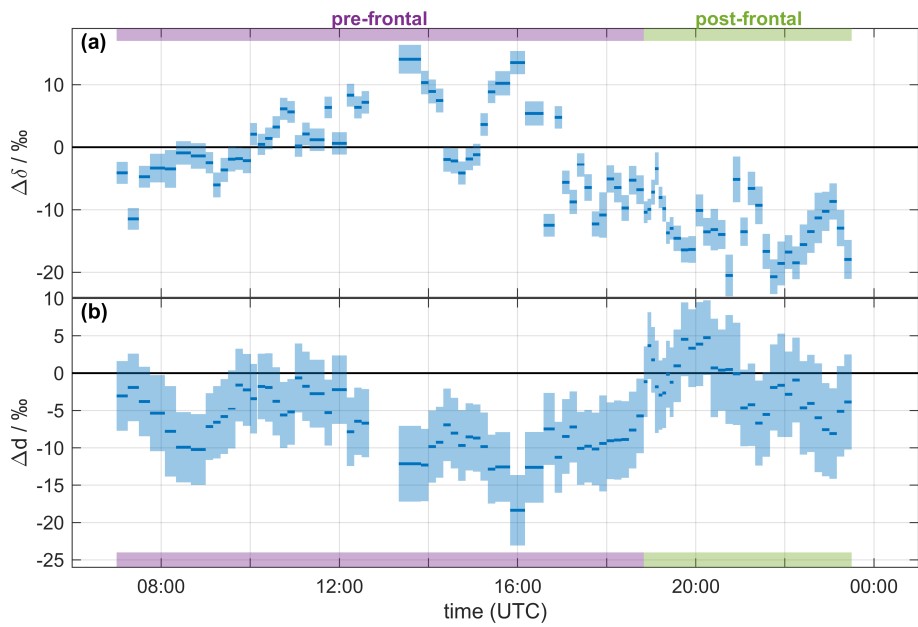

**Figure 3.** Time series of (a) $\Delta\delta$ and (b) $\Delta d$ of the precipitation samples collected on 20 November 2015. The width of the blue bars denotes the period over which the rain samples were collected. The calibrated uncertainties are indicated by the shaded areas. Pre- and post-frontal periods are indicated with purple and green bars, respectively.

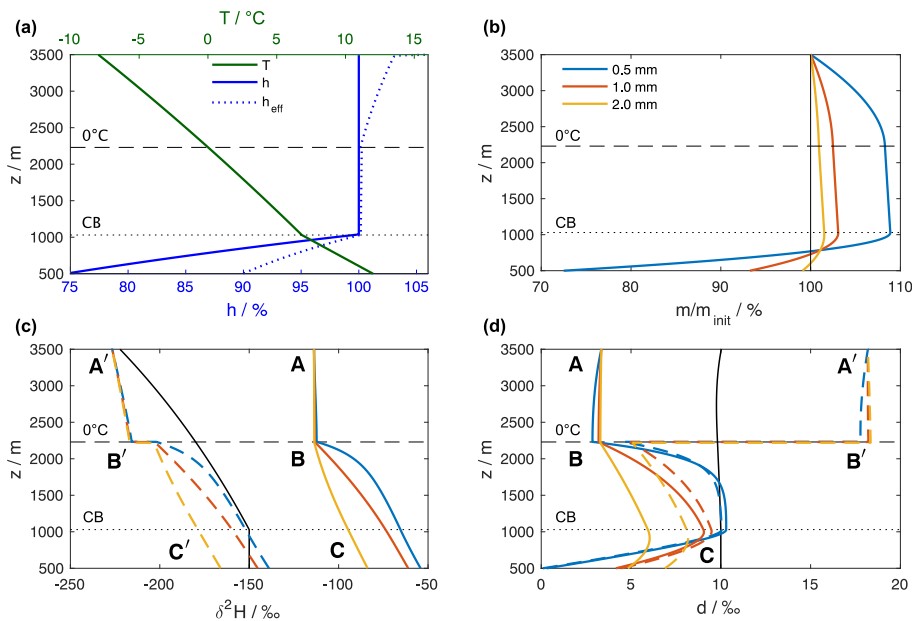

**Figure 4.** Results of the reference simulation of the single-column model. (a) Vertical profiles of air temperature (green line) and relative humidity over liquid (solid blue line), obtained from the (moist) adiabatic ascent of an air parcel from the surface with initial $T_0 = 12°C$ and $h_0 = 75\%$. The relative humidity of the surrounding air with respect to the temperature of the 1 mm hydrometeor, denoted as effective relative humidity $h_{eff}$, is shown as dotted blue line. (b) Hydrometeor mass relative to the initial mass at the formation height. The coloured lines correspond to 3 hydrometeors that arrive at the surface with an equivalent liquid diameter of 0.5, 1 and 2 mm, respectively. (c) Isotopic composition of hydrometeors and the surrounding vapour. Coloured lines show the isotopic composition of the hydrometeors ($\delta^2 H_p$, solid) and the equilibrium vapour from the hydrometeors ($\delta^2 H_{p,eq}$, dashed). The black line indicates the composition of the ambient vapour ($\delta^2 H_v$). The letters mark locations that are referenced in the text. (d) Same as (c) but for $d_p$ and $d_{p,eq}$. Horizontal dashed and dotted black lines in all plots mark the height of the 0°C-isotherm and the height of the cloud base (CB), respectively.

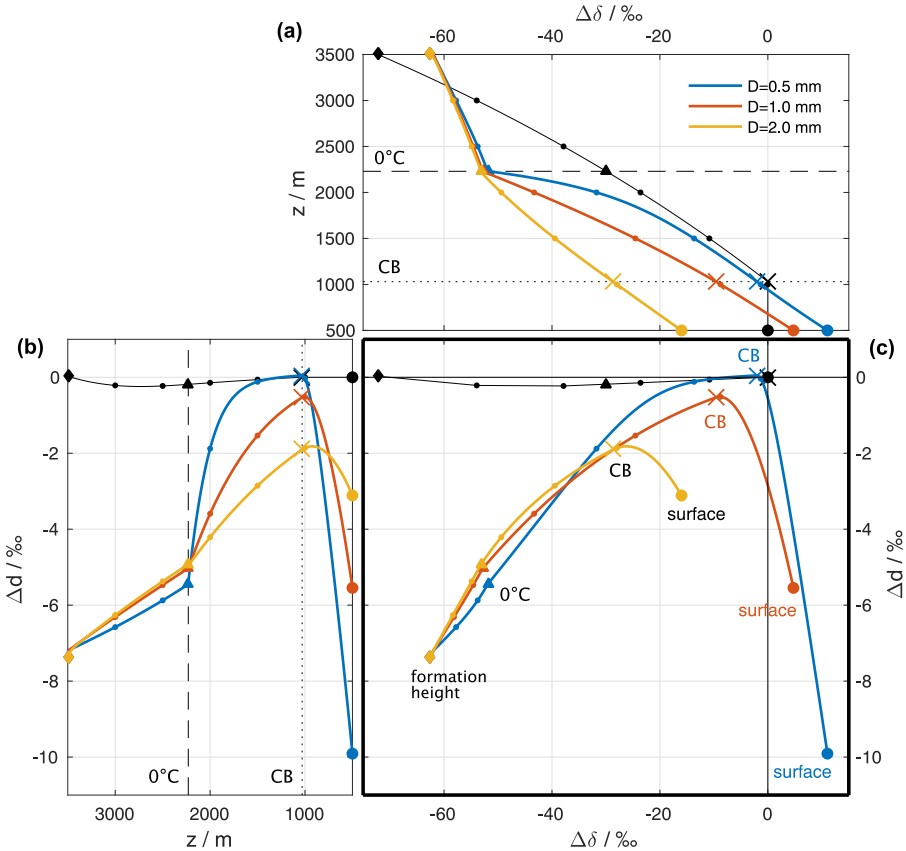

**Figure 5.** Isotopic composition of the hydrometeors and the surrounding vapour of a reference simulation (see Sec. 4). (a) Difference between the surface vapour and the equilibrium vapour from a falling liquid hydrometeor ($\Delta\delta$, coloured lines). The composition of the ambient vapour at different altitudes relative to surface vapour ($\delta^2 H_v - \delta^2 H_{v,0}$) is shown as black lines. The curves are similar to the coloured lines in Fig. 4c, but instead of the absolute value showing the deviation from the surface vapour composition. The horizontal dashed and dotted black lines mark the height of the $0°$C-isotherm and the height of the cloud base (CB), respectively. (b) same as (a), but for $d_p$ and $d_{p,eq}$ and rotated to match the y-axis of (c). (c) $\Delta\delta\Delta d$-diagram: $\Delta\delta$ from (a) vs. $\Delta d$ from (b). In all plots, the isotopic composition at every full 500 m is highlighted with a small dot. The compositions at the following altitudes are also highlighted: diamond: altitude of release; triangle: altitude of $T_d = 0°$C; cross: altitude of the cloud base; large filled circle: surface. For simplicity, the equilibrium vapour from a liquid hydrometeor is shown above the $0°$C-isotherm.

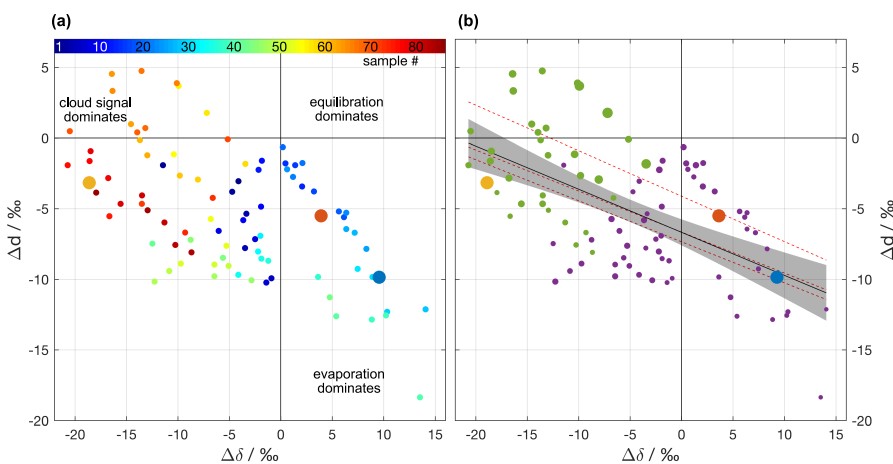

**Figure 6.** $\Delta\delta\Delta d$-diagram for the precipitation samples collected on 20 November 2015. (a) Samples coloured according to their sequential sample number (see legend) to highlight the temporal evolution of $\Delta\delta$ and $\Delta d$. (b) Same samples as in (a), but with pre-frontal samples coloured in purple and post-frontal ones in green. The size of the circle corresponds to the average rain intensity of the sample. The solid black line represents a linear fit through all samples with the 95% confidence band in shading. Dashed red lines correspond to the linear fits through the samples of 3 other events (cf. text). Data points from reference simulations shown as large yellow, red, and blue dots.

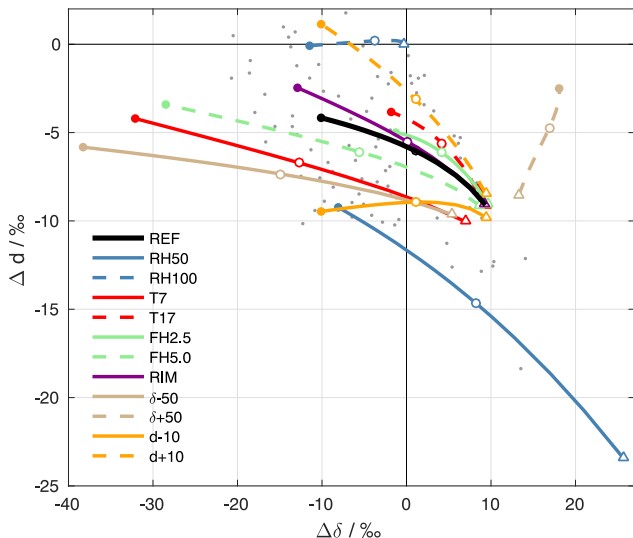

**Figure 7.** $\Delta\delta\Delta d$-diagram for the results of sensitivity experiments with the idealized below-cloud interaction model. The black line shows results from the reference setup, and coloured lines reveal the results from experiments with altered input parameters (see text for explanation). For each setup, a line is shown that connects results of simulations with different drop sizes (corresponding to surface precipitation intensities from 0.1 to 20 mm h$^{-1}$). The triangles correspond to the lowest rain intensity (0.1 mm h$^{-1}$). Empty circles correspond to an intensity of 2 mm h$^{-1}$ and filled circles correspond to an intensity of 20 mm h$^{-1}$. Grey dots show precipitation samples collected on 20 November 2015.

*Competing interests.* The authors declare that they have no conflict of interest.

*Data availability.* All observation data presented in this study are available from the authors upon request (heini.wernli@env.ethz.ch).

*Acknowledgements.* We acknowledge Franziska Aemisegger and Marina Dütsch for useful inputs and discussions concerning this work. We thank Patrick Bertollini for his help with precipitation sampling and analysis, Peter Isler for the meteorological data, Barbara Herbstritt (University of Freiburg) and Yongbiao Weng (University of Bergen) for reference measurements of our calibration standards. HS acknowledges funding from the Norwegian Research Council projects SNOWPACE (Project No. 262710) and FARLAB (Project No. 245907).

## Appendix A: Below-cloud interaction model

### A1 Initial conditions

Vertical profiles of pressure, temperature, humidity and the isotopic composition of water vapour are either manually defined (e.g., based on observations) or calculated from a (moist-)adiabatic ascent of an air parcel with given surface composition. The latter approach uses the Rayleigh model of Dütsch et al. (2017), with the difference that here a dry adiabatic ascent is allowed in the initial phase until saturation is reached at the cloud base. The air parcel starts at height $z_0$ with pressure $p_0$, temperature $T_0$ and relative humidity $h_0$ and ascends in height steps of $\Delta z$. The pressure $p_{k+1}$ and temperature $T_{k+1}$ at a higher level $z_{k+1} = z_k + \Delta z$ can be calculated using the lapse rate $\Gamma$ at level $z_k$:

$$T_{k+1} = T_k - \Gamma \Delta z \tag{A1}$$

$$p_{k+1} = p_k \left( \frac{T_{k+1}}{T_k} \right)^{\frac{g}{R_d \Gamma}} \tag{A2}$$

where for $\Gamma$ the dry adiabatic lapse rate $\Gamma_d$ is used for $h_k < 100\%$ or the moist adiabatic lapse rate $\Gamma_m$ for $h_k = 100\%$. They are defined as (Holton and Hakim, 2013):

$$\Gamma_d = \frac{g}{c_p} \tag{A3}$$

$$\Gamma_m = \frac{g}{c_p} \frac{1 + L_e w_k / (R_d T_k)}{1 + \epsilon L_e^2 w_k / (c_p R_d T_k^2)} \tag{A4}$$

where $g$ is the gravitational constant, $c_p$ is the specific heat of dry air at constant pressure, $w_k$ is the mass mixing ratio of water vapour in air at level $z_k$, $L_e$ is the latent heat of evaporation, $R_d$ is the specific gas constant of dry air, $T_k$ is the temperature at level $z_k$, and $\epsilon = 0.622$ is the ratio of the specific gas constants of dry air and water vapour.

All moisture above saturation condenses and immediately precipitates from the air parcel. $w_{k+1}$ can therefore be diagnosed as:

$$w_{k+1} = \begin{cases} w_k & \text{if } h_k < 100\% \\ \frac{\epsilon \cdot e_{\text{sat}}}{p_{k+1} - e_{\text{sat}}} & \text{if } h_k = 100\% \end{cases} \tag{A5}$$

The specific humidity $q_{k+1}$ can be calculated in both cases as:

$$q_{k+1} = \frac{w_{k+1}}{1 + w_{k+1}} \tag{A6}$$

$e_{\text{sat}}$ in equation (A5) is the saturation vapour pressure. It is defined as a combination of the saturation vapour pressures over liquid $e_{\text{sat}}^l$ and ice $e_{\text{sat}}^i$:

$$e_{\text{sat}} = f_c e_{\text{sat}}^l + (1 - f_c) S_i e_{\text{sat}}^i, \tag{A7}$$

where $f_c$ is the fraction of liquid water in the condensate, which changes from 0 to 1 as $T$ decreases. At $T > 0°C$, the condensate is purely liquid and hence $f_c = 1$ and $e_{\text{sat}} = e_{\text{sat}}^l$. Below $-23°C$ the condensate is purely frozen, $f_c = 0$ and $e_{\text{sat}} = e_{\text{sat}}^i \cdot S_i$.

The supersaturation with respect to ice $S_i$ takes the form $S_i = 1 - \lambda T$ for $T < -23°C$ with $\lambda = 0.004$ (see Risi et al., 2010b) and $S_i = e_{\text{sat}}^l / e_{\text{sat}}^i = 1$ for $T = 0°C$. For $-23°C < T < 0°C$, both $f_c$ and $S_i$ are interpolated cubically.

For the parameterization of isotopic fractionation, the Rayleigh model follows Merlivat and Jouzel (1979) for liquid clouds and Jouzel and Merlivat (1984) for solid clouds, with the difference that both solid and liquid condensates are immediately removed from the air parcel (see again Dütsch et al., 2017). The isotope ratios $R_{k+1}$ in water vapour at height level $z_{k+1}$ is calculated from the Rayleigh distillation equation:

$$R_{k+1} = R_k \left( \frac{q_{k+1}}{q_k} \right)^{\alpha_{\text{eff}} - 1}, \tag{A8}$$

where $\alpha_{\text{eff}}$ is the effective fractionation factor that it is defined as a combination of the equilibrium fractionation factors with respect to liquid ($\alpha_{v \rightarrow l}$) and ice ($\alpha_{v \rightarrow s}$), depending on the type of condensate that is formed, and the nonequilibrium fractionation factor in mixed-phase clouds $\alpha_k$:

$$\alpha_{\text{eff}} = f_c \alpha_{v \rightarrow l} + (1 - f_c) \alpha_{v \rightarrow s} \alpha_k \tag{A9}$$

$$\text{with} \quad \alpha_k = \frac{S_i}{\alpha_{v \rightarrow s} \mathcal{D}/\mathcal{D}'(S_i - 1) + 1} \quad \text{(Jouzel and Merlivat, 1984)}, \tag{A10}$$

where $\mathcal{D}/\mathcal{D}'$ is the ratio of the diffusion coefficients of the light and heavy isotopes (taken from Merlivat (1978)), and $\alpha_{v \rightarrow l}$ and $\alpha_{v \rightarrow s}$ are specified following Majoube (1971b), Majoube (1971a) and Merlivat and Nief (1967).

The resulting vapour profile has a constant isotopic composition below the cloud base, where the ascending air parcel reaches saturation. Above the cloud base, $q$ decreases with height, and the air parcel preferentially loses heavy isotopes that are precipitated, resulting in a profile that is increasingly depleted with height in $\delta^2 H_v$ and $\delta^{18} O_v$. The evolution of $d_v$ depends on the initial isotopic composition of the vapour at the surface.

## A2 Initial isotopic composition of the hydrometeor

Hydrometeors in the model are released at a given altitude. They are assumed to be formed entirely from the vapour at this altitude and their initial composition is calculated from the isotopic composition of the surrounding vapour at ambient temperature. This is a strong simplification, as in reality hydrometeors grow over time, accumulating mass from different altitudes. The formation mechanism has a large influence on the initial isotopic composition of a hydrometeor, in particular on its deuterium excess, because nonequilibrium effects may be involved. Two different formation mechanisms that are important for mid-latitude precipitation have been implemented: Growth by vapour deposition in mixed-phase or ice clouds, and riming of liquid cloud droplets on frozen particles. Direct freezing of liquid hydrometeors, which occurs in strong updrafts and at low temperature, corresponds to the case of a purely rimed hydrometeor.

### A2.1 Growth by vapour deposition

Precipitation formation in both mixed-phase and ice clouds occurs by deposition of vapour on ice particles. Ice particles and liquid cloud droplets coexist for $-23°C < T < 0°C$ and the cloud is assumed to be entirely composed of ice below $-23°C$.

As desribed in section A1, non-equilibrium fractionation due to supersaturation with respect to ice is taken into account with a kinetic fractionation factor $\alpha_k$ (Eq. A10):

$$R_{p,z_{\text{start}}} = \alpha_{v \to s} \alpha_k R_{v,z_{\text{start}}} \tag{A11}$$

where $R_{v,z_{\text{start}}}$ is the isotopic composition of ambient vapour at the starting height.

## A2.2 Riming

If ice particles and supercooled water droplets coexist in mixed-phase clouds, the icea particles may grow by colliding with freezing droplets, a process called riming. Riming is favoured when ice particles fall through a supercooled liquid cloud and the concentration of cloud droplets is high, e.g., in strong updrafts. No fractionation occurs during riming, because the entire droplet is rapidly transformed to ice. A rimed hydrometeor consists of a mass fraction formed by vapour deposition (see above) and a fraction formed by contact freezing of supercooled cloud droplets. The resulting initial composition of a rimed hydrometeor can therefore be described as a combination:

$$R_{p,z_{\text{start}}} = ((1 - f_r)\alpha_{v \to s}\alpha_k + f_r\alpha_{v \to l}) \cdot R_{v,z_{\text{start}}} \tag{A12}$$

where $f_r$ is the rimed mass fraction. It varies between 0 for no riming and 1 for riming only and can be adjusted in the model to test the sensitivity to the rimed mass fraction. Here, similar to Blossey et al. (2010), the composition of supercooled liquid cloud droplets in a mixed-phase cloud is assumed to be in equilibrium with the vapour phase ($R_l = \alpha_{v \to l} \cdot R_v$), as if no ice particles were present, neglecting the additional non-equilibrium fractionation that occurs in reality. More exact formulations (Ciais and Jouzel, 1994; Bolot et al., 2013), however, depend on variables that are not included in our model, e.g., the precipitation rate of solid hydrometeors, and would require additional assumptions.

## A3 Microphysics of a falling hydrometeor

Mass and temperature of the hydrometeor are calculated along its fall trajectory using a sufficiently small time step $\Delta t = 0.1\,\text{s}$ to avoid numerical instability. The hydrometeor is assumed to fall at the terminal velocity $v_T$ of a liquid drop of diameter $D$ immediately after release. $v_T$ is calculated following Foote and Du Toit (1969), including corrections for the aspherical shape of large drops and the lower air density aloft ($\rho$):

$$v_T = -9.43\,\text{m\,s}^{-1}\left\{1 - \exp\left(-\frac{D}{1.77\,\text{mm}}^{1.147}\right)\right\}\left(\frac{\rho_0}{\rho}\right)^{0.4} \tag{A13}$$

where $v_T$ is negative for a falling hydrometeor and $\rho_0 = 1.2038\,\text{kg\,m}^{-3}$ is a reference air density. The terminal velocity of frozen particles, especially snow flakes, can be much smaller, and this allows more time to grow, shrink or exchange isotopes. However, shrinking and exchanging isotopes are non-fractionating processes for solid hydrometeors and the effect of growth on the isotopic composition of the particle is small for our application. Therefore, an explicit formulation of the fall velocity for solid particles is omitted for simplicity.

To calculate the change of mass and temperature between $z_t$ and $z_{t+1}$, the environmental values of temperature, pressure and humidity are interpolated between the two heights. The change of mass of a falling hydrometeor is calculated as (Pruppacher and Klett, 2010)

$$\frac{dm}{dt} = \frac{2\pi D f_v \mathcal{D}}{R_w^*} \left( h \frac{e_{\mathrm{sat}}(T_\infty)}{T_\infty} - \frac{e_{\mathrm{sat}}(T_d)}{T_d} \right) \tag{A14}$$

with hydrometeor diameter $D$, ventilation coefficient $f_v$, diffusivity of vapour $\mathcal{D}$, specific gas constant for water vapour $R_w^* = R/M_w$, ambient relative humidity $h$, ambient temperature $T_\infty$, hydrometeor temperature $T_d$, saturation vapour pressure in the ambient air $e_{\mathrm{sat}}(T_\infty)$, and saturation vapour pressure above the hydrometeor surface $e_{\mathrm{sat}}(T_d)$.

$T_d$ has to be determined before calculating $\frac{dm}{dt}$. This is done by considering the heat balance of the hydrometeor, which is given by

$$\frac{dQ}{dt} = L_e \frac{dm}{dt} + \frac{dh_s}{dt} \tag{A15}$$

where $L_e$ is the latent heat of evaporation. The first term is defined as the loss of heat due to evaporation and can be calculated by substituting equation (A14). The second term is the sensible heat transferred to or from the environment and can be substituted by the heat transfer equation (Abraham, 1968):

$$\frac{dh_s}{dt} = -2\pi D f_h k_a (T_d - T_\infty) \tag{A16}$$

where $f_h$ is the heat ventilation coefficient and $k_a$ is the thermal conductivity of air. Expressing $dQ/dt$ in Eq. (A15) as

$$\frac{dQ}{dt} = m c_w \frac{dT_d}{dt} = \frac{1}{6} \pi D^3 \rho_w c_w \frac{dT_d}{dt} \tag{A17}$$

leads to the following formula for the temperature change of a falling hydrometeor (Salamalikis et al., 2016):

$$\frac{dT_d}{dt} = \frac{12}{D^2 \rho_w c_w} \left\{ \frac{L_e f_v \mathcal{D}}{R_w^*} \left( h \frac{e_{\mathrm{sat}}(T_\infty)}{T_\infty} - \frac{e_{\mathrm{sat}}(T_d)}{T_d} \right) - f_h k_a (T_d - T_\infty) \right\}, \tag{A18}$$

where $c_w$ is the specific heat of liquid water. If the relative humidity $h$ is below a certain value, the term $\left( h \frac{e_{\mathrm{sat}}(T_\infty)}{T_\infty} - \frac{e_{\mathrm{sat}}(T_d)}{T_d} \right)$

becomes negative and the hydrometeor starts to evaporate. Evaporation will decrease the temperature of a falling hydrometeor. Neglecting this cooling would lead to an overestimation of evaporation.

## A4   Calculation of the isotopic composition along the fall trajectory of a hydrometeor

The isotopic composition of the falling hydrometeor can be diagnosed from the time evolution of its temperature $T_d$ and mass $m$. Following, e.g., Pfahl et al. (2012), we can calculate a mass balance for each isotopic species individually. The mass

tendency of each species is related to the total mass tendency $\frac{dm}{dt}$ by

$$\frac{dm'}{dt} = \frac{dm}{dt} \left( \frac{f_v' \mathcal{D}'}{f_v \mathcal{D}} \right)^n \frac{h \cdot \rho_{\mathrm{sat}}'(T_\infty) - \rho_{\mathrm{sat}}'(T_d)}{h \cdot \rho_{\mathrm{sat}}(T_\infty) - \rho_{\mathrm{sat}}(T_d)}, \tag{A19}$$

where the quantities associated with heavy isotopic species are denoted with a prime. The exponent n is chosen to be 0.58, based on the measurements by Stewart (1975). It corrects for the fact that isotopic transport between vapour above the hydrometeor

surface and the environment is not purely diffusional for a falling hydrometeor, but involves turbulence. Inserting Eq. (A14) into Eq. (A19), and using $\rho_{\text{sat}}(T) = \frac{e_{\text{sat}}(T)}{R_w^* T}$ results in the following expression:

$$\frac{dm'}{dt} = \frac{2\pi D f_v \mathcal{D}}{R_w^*} \left(\frac{f_v' \mathcal{D}'}{f_v \mathcal{D}}\right)^n \left(h \cdot \frac{e_{\text{sat}}'(T_\infty)}{T_\infty} - \frac{e_{\text{sat}}'(T_d)}{T_d}\right) \tag{A20}$$

where $e_{\text{sat}}'(T_\infty) = R_v \cdot e_{\text{sat}}(T_\infty)$ and $e_{\text{sat}}'(T_d) = \frac{R_p}{\alpha_{v \to l/s}} \cdot e_{\text{sat}}(T_d)$. The expressions for $f_v'$ and $\mathcal{D}'$ can be found in Graf (2017).

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
