# Peer review of "A new interpretative framework for below-cloud effects on stable water isotopes in vapour and rain"

_Atmospheric Chemistry and Physics, 2018_

## Referee Comment (RC1) · Anonymous Referee #1 · 28 Jun 2018

Review results for Graf et al. "A new interpretative framework for below-cloud effects on stable water isotopes in vapour and rain"

In this research, the authors analyzed stable isotope ratio timeseries of a rain event in Switzerland to investigate below-cloud hydrological processes. Importance of below-cloud processes has been increased because of emergence of high-resolution cloud resolving atmospheric model, for example. A new approach to use both rain and vapor isotope ratio timeseries is proposed. In this approach, anomaly of the observed surface vapor isotope ratio is subtracted from equilibrium isotope ratio from the observed precipitation isotope ratio. Then, according to the authors, only the effect of below-cloud processes is extracted.

The approach seems indeed interesting and novel. However, there is significant ig-

[Figure]

norance of the theory of kinetic fractionation. In case of unsaturated condition, there is always kinetic fractionation occurring during either evaporation or isotopic exchange (e.g., Craig and Gordon 1965, Merlivat and Jouzel, 1978, Stewart 1975, etc.). Equilibrium fractionation is defined in case of saturated condition. In the present paper, the situation of $\Delta\delta=\Delta d=0$ occurred when RH<100%. That means, the situation did not happen because the vapor and liquid were in the equilibrium state. Rather than that, it was occurred by kinetic fractionation process depending on specific RH and the initial dD and d18O values of vapor and liquid.

The kinetic fractionation would behave quite complicatedly in $\Delta\delta\Delta d$ diagram, according to Stewart's (1975) formulation, which is the most popular parameterization in the isotope general circulation models, for example, $\Delta\delta$ and $\Delta d$ are highly sensitive to the initial isotopic values of rain and vapor and RH of the ambient air. On the other hand, "the degree of equilibration" would not make such a big difference in $\Delta\delta\Delta d$ diagram. As stated above, there is always kinetic fractionation occurring in unsaturated condition, so if such kinetic fractionation's final state is practically called "equilibrium" (by the way, this is what is parameterized in the most of the models, e.g., Hoffmann et al., 1998; Yoshimura et al., 2008), this equilibration would not always become $\Delta\delta=\Delta d=0$, because there is kinetic fractionation process going on.

This ignorance of kinetic fractionation processes significantly influences the interpretation of the paper. For example, P9L17 "Strongly equilibrated sample are thus located close to the origin" is probably misleading. That is true when RH=100%, but not true when RH<100%. P9L21 "Rain samples that area strongly affected by evaporation will thus be located in the bottom right quadrant" may be misleading too. It is highly depended on initial condition and RH, and different initial condition and RH may cause different trend in evaporation line in $\Delta\delta\Delta d$. So, the right bottom position would not be reflected by "stronger evaporation".

By these reasons, I'd recommend the editor to reject the manuscript ang give them plenty time for resubmission.

Minor issues follow: P2L16: Did the authors come across any new understanding of below-cloud processes? P8L8: What is "cloud signal"? P8L12: More explanation for the other three events are necessary. Another big issue of the paper is lack of observation data. Are the characteristic of cold fronts similar? How about temporal tendency of the evaporation strength? P8L16: "less affected by blow-cloud processes": What does it mean? Isn't it contradicted from "stronger cloud process" in L8?
* * *

---

## Referee Comment (RC2) · Anonymous Referee #2 · 25 Jul 2018

**General Comments**

In this paper, the authors present a high-resolution time series of water isotope ratios in vapor and precipitation during a rain event in Switzerland. The goal of analyzing both together is to try and improve the understanding of what happens to rain as it falls through the convective boundary layer, where it is subject to a number of processes which can be difficult to observe. The $\Delta\delta\Delta d$ approach highlighted here is interesting and novel. However, there are some aspects of the paper which need to be clarified better (see specific comments below) and more extensive discussion is needed for the effects of non-equilibrium kinetic fractionation, which is a big gap in the current analysis. While the proposed viewing diagram is novel and would be useful if appropriate, care must be taken into reading too much into the information presented as the current analysis/interpretation is missing some key discussion of how kinetic fractionation would be represented on the $\Delta\delta\Delta d$ diagram. In addition, the definition of the $\Delta\delta$ and $\Delta d$ variables hinges on calculating the vapor-equivalent of rain using surface temperature. Ambient temperature is more likely to reflect the ambient vapor, and therefore I think the analysis should be done using the precipitation-equivalent of vapor and comparing it to the rain samples. This may not make a difference, but should be checked given the non-linearity brought up by the authors and the fact that all conclusions are based on this definition.
I would suggest returning the manuscript to the authors for revisions.

| Principal criteria | Excellent (1) | Good (2) | Fair (3) | Poor (4) |
|---|---|---|---|---|
| **Scientific significance:**

 Does the manuscript represent a substantial contribution to scientific progress within the scope of Atmospheric Chemistry and Physics (substantial new concepts, ideas, methods, or data)? | | 2 | | |
| **Scientific quality:**

 Are the scientific approach and applied methods valid? Are the results discussed in an appropriate and balanced way (consideration of related work, including appropriate references)? | | | 3 | |
| **Presentation quality:**

 Are the scientific results and conclusions presented in a clear, concise, and well-structured way (number and quality of figures/tables, appropriate use of English language)? | | 2 | | |

**Specific Comments**

**Abstract**
L11: Does 'equilibration' refer to 'exchange between ambient air and raindrops' or 'temperature-dependent equilibrium fractionation'? The description of the $\Delta\delta\Delta d$ figure in the previous sentence highlights the latter, but I tend to associate equilibration with the former process. L15 also cites equilibration being less when RH is higher – RH would not affect the temperature-dependent equilibrium but would affect the exchange of isotopes between rain and ambient air through rain evaporation and condensation. See Section 2 of Nusbaumer et al

(2017) for a good description of the microphysical processes needed to describe isotope ratios in rain and vapor ("Evaluating hydrological processes in the Community Atmosphere Model Version 5 (CAM5) using stable isotope ratios of water", *Journal of Advances in Modeling Earth Systems, 9:949-977*). Non-equilibrium kinetic fractionation must also be included in this analysis.

**Section 1**
P2, L30: It might help to insert some explanation about Rayleigh distillation along rain back trajectory, which is the starting point for getting to more depleted rain values at higher latitudes.
P2, L31: as long as the air column is unsaturated
P3, para1: Some introduction about kinetic fractionation would be appropriate here.

**Section 2**
P4, L2: How much of the data was discarded for both measurements and calibrations? This is typically done to ensure no memory effects on the final values.
P4, L9: Was any oil added to the funnel to prevent evaporation during collection? Did the authors check for potential sample evaporation over 30 minutes, especially if the air was unsaturated? This could especially complicate the interpretation of *d*-excess values.
P5, L4: This is also complicated by needing to know both temperature of vapor and rain.
P5, L15: Is ambient temperature accurately reflecting raindrop temperature? Wouldn't it be better to look at *"equilibrium precipitation from vapor"*, since ambient temperature is more likely to represent the ambient vapor? I would suggest repeating the analysis using this direction (or at least check to see if it makes a difference). Especially given the authors' explanation for non-linearity in temperature control of isotope ratios.
P5, L24: Do you get the same results using $\delta^{18}O$ observations instead for eqn (5)?

**Section 3**
P7, L19: I assume the correlations reported here are significant with p values < 0.05? Could add 'significant' to line 19. Correlation with *h* has been seen by others (Crawford et al 2016), especially in unsaturated semi-arid environments. Correlation with rain intensity/amount is traditionally assigned to the classic 'isotopic amount effect'. Some discussion of these would be appropriate here.
P7, L24: Consider: is the 'surrounding vapor' simply the vapor coming down with the rain in a downdraft? In which case I would expect them to be closely correlated. Can you distinguish this from non-downdraft/near-surface surface vapor being influenced by rain?

**Section 4**
Fig 3 and P7, L30: how do you distinguish 'rain samples in equilibrium with vapor' – is this the zero line? I don't see samples at 17 UTC near the zero line for $\Delta\delta$ or for $\Delta d$ at 21 UTC (looks more like 22 UTC). There are several periods where the uncertainty of $\Delta d$ overlaps the zero line.
P7, L32-33: More negative values of $\Delta\delta$ are seen before the frontal passage, with $\Delta d$ trending towards zero. Some explanation of what is going on here? Also, if there is conservation of the depleted isotopic signature from the cloud, I think this should also be reflected in $\Delta d$ being

zero. Any partial/incomplete tendency towards equilibration should be seen in both signatures from the way equations (5) and (6) are set up. If the argument is for 'conservation', i.e. vapor reflecting the original rain signature in equilibrium (possibly because it is associated with a downdraft associated with the frontal passage), this should be captured in both $\Delta\delta$ and $\Delta d$.
Fig 4, P8 L6-7: Some discussion of how non-equilibrium kinetic effects would influence the evolution of the $\Delta\delta$ and $\Delta d$ signals would be appropriate here. Why would the early pre-frontal samples be expected to be around the (0,0) point? Falling through an unsaturated atmosphere at 75-85% humidity, I would expect to see more evaporation and kinetic effects. Fig 4a is easier to interpret, the various transitions during the progression of the rain event are harder to read in Fig 4b.

**Section 5**
P9, para 2: Here the discussion of $\Delta\delta$ and $\Delta d$ is in terms of precipitation, while previously it was defined as the difference in precipitation-equilibrated and surface vapors.
P9, L11: Assuming the precipitation is in equilibrium with the vapor it formed from in the cloud (through temperature equilibration), I would expect the $\Delta d$ of rain versus 'vapor-where-the-rain-formed-in-the-cloud' to be small, but it could still be vastly different from the $\Delta d$ at the surface.
P9, L15: Raindrops will also get more enriched during evaporation into an unsaturated atmosphere, because the lighter isotopes are preferentially evaporated, which will make the ambient vapor more depleted.

**Technical Corrections**

**Abstract**
L4: delete 'to'
L17: either 'this type of rain event' or 'these types of rain events'.

**Section 1**
P2, L6: 'but do neither provide' -> 'but provide neither'
P2, L14: Refs in order of date or alphabetical?
P2, Eqn (2): should be a x1000 in there
P3, L15: delete 'a'
P3, L24: 'allows to' -> 'allows us to'

**Section 3**
P6, L5: 'extends' -> 'extended'
Fig 1 caption: 'measurement site at Zurich'

---

## Author Comment (AC1) · 4 Nov 2018

**Reply to Referee Comments for manuscript acp-2018-482 by Pascal Graf, Heini Wernli, and Harald Sodemann: A new interpretative framework for below-cloud effects on stable water isotopes in vapour and rain**

**Comments of the referees are in *italics*; our replies are in normal font.**

We thank both reviewers for their comments, which helped to clarify the manuscript. The originally submitted version lacked a clear explanation of the effects of kinetic fractionation and its influence on $\Delta\delta$ and $\Delta d$. Thus, in our substantially revised version we decided to include and discuss the results of idealised simulations with a below-cloud interaction model to illustrate the influence of different atmospheric parameters on the location of precipitation samples in the $\Delta\delta\Delta d$-space. As a consequence, we include Stephan Pfahl as a co-author in the revised manuscript.

**Comments of referee 1**

**Answers on general comments:**

1. *There is significant ignorance of the theory of kinetic fractionation. In case of unsaturated condition, there is always kinetic fractionation occurring during either evaporation or isotopic exchange. Equilibrium fractionation is defined in case of saturated condition.*

REPLY: We did not ignore kinetic fractionation. While there are several passages in the text where this is highlighted (P3, L.1; P8, L.10, P8, L.17, P8, L. 20), we acknowledge that we did not mention its influence clearly enough, in particular on $\Delta\delta$ and $\Delta d$. We outline below how kinetic fractionation is related to $\Delta\delta$ and $\Delta d$, and improve the relevant descriptions in the manuscript:

We agree that kinetic fractionation occurs in unsaturated conditions. In addition, kinetic fractionation occurs in saturated conditions when vapour and precipitation are far from their equilibrium composition, i.e., in the presence of large gradients for a single isotope species. The latter case is however usually ignored when qualitatively interpreting isotope data because the effects are small. Equilibrium fractionation in contrast occurs during both saturated and unsaturated conditions. The total fractionation during unsaturated conditions is then a combination of equilibrium and kinetic/non-equilibrium fractionation. We define $\Delta\delta$ and $\Delta d$ as the difference between ambient vapour and equilibrium vapour of precipitation. Under saturated conditions, given enough time, this difference becomes 0‰ due to equilibration. In unsaturated conditions, equilibration also acts to reduce $\Delta\delta$ and $\Delta d$, it is however counteracted and often outweighed by the effect of non-equilibrium fractionation, which increases $\Delta\delta$ and decreases $\Delta d$.

2. *In the present paper, the situation of $\Delta\delta=\Delta d=0$ occurred when RH<100%. That means, the situation did not happen because the vapour and liquid were in equilibrium state. Rather than that, it was occurred by kinetic fractionation process depending on specific RH and the initial dD and d18O values of vapour and liquid.*

REPLY: The situation $\Delta\delta=\Delta d=0$‰ at RH<100% indeed does not occur when vapour and rain have enough time to interact. Evaporation will increase $\Delta\delta$ and decrease $\Delta d$ and, if strong compared to equilibration, move the rain sample away from $\Delta\delta=\Delta d=0$‰. Thus, when falling through unsaturated air, precipitation must have had a negative $\Delta\delta$ and a positive $\Delta d$ in order to arrive at the surface at $\Delta\delta=\Delta d=0$‰. We agree that the situation probably did not occur because they were in a state of equilibrium. However, arriving at the surface with $\Delta\delta=\Delta d=0$‰ excludes strong evaporation, for which a strongly positive initial $\Delta d$ would be necessary, which is not realistic (cf. P9, L11-14). More realistic is either a saturated column, except for the lowermost layer, where RH was measured, or precipitation obtained a positive $\Delta d$ by equilibration with vapour aloft, with a higher $d_v$ than vapour at the surface.

3. *The kinetic fractionation would behave quite complicatedly in $\Delta\delta\Delta d$ diagram, according to Stewart's (1975) formulation, which is the most popular parameterization in the isotope general circulation models, for example, $\Delta\delta$ and $\Delta d$ are highly sensitive to the initial isotopic values of rain and vapor and RH of the ambient air. On the other hand, "the degree of equilibration" would not make such a big difference in $\Delta\delta\Delta d$ diagram. As stated above, there is always kinetic fractionation occurring in unsaturated condition, so if such*

*kinetic fractionation's final state is practically called "equilibrium" (by the way, this is what is parameterized in the most of the models, e.g., Hoffmann et al., 1998; Yoshimura et al., 2008), this equilibration would not always become Δδ=Δd=0, because there is kinetic fractionation process going on.*

REPLY: We want to first clarify how we use the term equilibration. By equilibration we mean the exchange of isotopes between vapour and rain in saturated air. In unsaturated air, we assume that this equilibration is the same and acts along the same direction, but it is complemented by evaporation, which makes the final trajectory of a sample in the ΔδΔd-space different. The final trajectory is the sum of the vectors of pure equilibration and evaporation (the length of the vectors depends on the intensity of the processes). This combination would not always cause samples to move to Δδ=Δd=0‰, as you state correctly, it would move them to 'kinetic fractionations final state' or what Stewart (1975) described with $\delta_{end}$ (his Eq. 3c).

Kinetic fractionation is part of the evaporation process and acts along a well-defined direction in the ΔδΔd-space when separated from the equilibration process, i.e., when RH=0%. Using Stewart's approach, the slope of the direction along which evaporation acts is around –1.3 +/– 0.4 Δd/Δδ with dependencies on the delta-value, temperature and on which fraction of the initial drop is remaining (see Fig. A in this reply document). The difference for varying conditions is thus not very large. Evaporation always acts towards the bottom right of the ΔδΔd-diagram. Equilibration without evaporation (RH=100%) however, does indeed depend strongly on the isotopic difference between equilibrium vapour from precipitation and vapour ('local' Δδ and Δd, see answer to P2 of reviewer 2). At the surface, equilibration acts towards the origin (0,0), denoting the composition of near-surface vapour in the ΔδΔd-diagram.

Thus, the trajectories of pure equilibration and evaporation act in very different directions in the ΔδΔd-space, which helps to separate their influence. The final location of a sample in the ΔδΔd-diagram is then an interplay of the initial composition and its modification by evaporation and equilibration. Since we can constrain the initial composition by a certain extent (P9, L10 onwards), and because we know the directions into which equilibration and evaporation act, we can deduce information about the relative importance of below-cloud processes from the location of a sample in the ΔδΔd-space. In the revised version, we will add a discussion of these factors to Sec. 5 and the Conclusions, using also results from the idealized simulations with the below-cloud interaction model.

4. *This ignorance of kinetic fractionation processes significantly influences the interpretation of the paper. For example, P9L17 "Strongly equilibrated sample are thus located close to the origin" is probably misleading. That is true when RH=100%, but not true when RH<100%. P9L21 "Rain samples that are strongly affected by evaporation will thus be located in the bottom right quadrant" may be misleading too. It is highly depended on initial condition and RH, and different initial condition and RH may cause different trend in evaporation line in ΔδΔd. So, the right bottom position would not be reflected by "stronger evaporation".*

REPLY: We agree that the formulation can appear too strong and lead to misinterpretation. We rephrased these sentences as follows: Strongly equilibrated samples *tend* to be located close to the origin. They can also be there due to other reasons, such as specific initial conditions. However, initial conditions have a limited range of realistic values. Samples without evaporation are therefore unlikely to be located in the bottom right corner. Samples from highly intense rain and a low melting layer will not be located close to the origin and samples from very light rain and high saturation will not be located in the bottom right or on the left of the ΔδΔd-diagram, because the initial conditions necessary for this to happen are not realistic. In the revised manuscript, we include a set of sensitivity experiments with a below-cloud fractionation model to explore a range of realistic variation in the ΔδΔd-space.

**Answers to specific comments:**

*P2L16: Did the authors come across any new understanding of below-cloud processes?*

REPLY: We show that by means of a new interpretation framework for below-cloud processes using stable isotopes, the effect of equilibration and evaporation on precipitation below the cloud can (to some extent) be separated. In addition, we identify the role of key factors during a frontal passage, including boundary-layer humidity, precipitation intensity and melting layer height. As an example, this was stated in item 4 of the

conclusions as "Post-frontal samples are less equilibrated and evaporated than pre-frontal samples, due to higher below-cloud relative humidity and a lower temperature and melting layer after the frontal passage." Our study paves the ground for further work which can employ this framework in different weather situations. We will emphasize these aspects even more clearly in the abstract and conclusions of the revised manuscript.

*P8L8: What is "cloud signal"?*

REPLY: By 'cloud signal' we refer to the initial composition of precipitation after formation, which is governed by the isotopic composition of the cloud vapour (amongst other influences like the temperature during formation and the formation mechanism). We adapted the text in order to clarify:
"Data points to the left of the origin indicate that precipitation is more depleted than ambient air, and reflect that more of the initial signal after formation ("cloud signal") is retained in precipitation"

*P8L12: More explanation for the other three events are necessary. Another big issue of the paper is lack of observation data. Are the characteristic of cold fronts similar? How about temporal tendency of the evaporation strength?*

REPLY: The other three events are presented in detail in Graf (2018). Including them in this manuscript with proper consideration of the measurement setup, data treatment, meteorological description and interpretation would go beyond its scope. We do not understand the comment about a lack of observations data, given that a unique detailed data set of the frontal transition in terms of high-resolution vapour and precipitation isotope measurements with 86 samples are used in this manuscript.

*P8L16: "less affected by blow-cloud processes": What does it mean? Isn't it contradicted from "stronger cloud process" in L8*

REPLY: We distinguish two main influences on the isotopic signal of surface precipitation (cf. P9L4+5): the initial composition after formation ("cloud signal") and its alteration by below-cloud processes (also called post-condensation processes). If precipitation is 'less affected by below-cloud processes', it carries more of the 'cloud signal' to the surface. If it is 'strongly affected by below-cloud processes', the 'cloud signal' is overwritten by below-cloud processes on the way to the surface. The *influence* of the 'cloud signal' thus depends on the strength of below-cloud processes and not on the *strength* of the 'cloud signal', (referred to as 'cloud processes' by the reviewer).

**Comments of referee 2**

**Answers on specific comments:**

**Abstract**
*L11: Does 'equilibration' refer to 'exchange between ambient air and raindrops' or 'temperature-dependent equilibrium fractionation'? The description of the $\Delta\delta\Delta d$ figure in the previous sentence highlights the latter, but I tend to associate equilibration with the former process. L15 also cites equilibration being less when RH is higher – RH would not affect the temperature-dependent equilibrium but would affect the exchange of isotopes between rain and ambient air through rain evaporation and condensation. See Section 2 of Nusbaumer et al (2017) for a good description of the microphysical processes needed to describe isotope ratios in rain and vapor ("Evaluating hydrological processes in the Community Atmosphere Model Version 5 (CAM5) using stable isotope ratios of water", Journal of Advances in Modeling Earth Systems, 9:949-977). Non-equilibrium kinetic fractionation must also be included in this analysis.*

REPLY: 'Equilibration' refers to 'exchange between ambient air and raindrops'. By including the 'equilibrium vapour from precipitation samples' ($\delta^2H_{p,eq}$, $d_{p,eq}$) instead of the actual composition of precipitation samples ($\delta^2H_p$, $d_p$) in Eqs. (5) and (6), we remove the 'equilibrium difference' (cf. P5,L1), which is caused by the 'temperature-dependent isotopic fractionation'. The high relative humidity reduces below-cloud evaporation and the low melting layer reduces time for equilibration and thus affects the 'degree of

equilibration' between rain and vapour on the ground. We agree that the formulation in L15 is somewhat unclear and could be confusing. We therefore rephrased the sentence in the revised manuscript: "After the frontal passage, the near-surface atmospheric layer is characterised by higher relative humidity, which leads to weaker below-cloud evaporation. Additionally, a lower melting layer after the frontal passage reduces time for exchange between vapour and rain and leads to weaker equilibration."

**Section 1**

*P2, L30: It might help to insert some explanation about Rayleigh distillation along rain back trajectory, which is the starting point for getting to more depleted rain values at higher latitudes.*

REPLY: We extended the sentence by the explanation that air at high altitudes (and latitudes) has experienced more rainout and is thus more depleted due to the Rayleigh distillation process. "As heavy isotopes preferentially condense due to their larger mass, air subject to rainout subsequently loses heavy isotopes. The increasing depletion with increasing fraction of rainout along the trajectory of an air parcel can be approximated by the Rayleigh distillation model under most atmospheric conditions (Dansgaard, 1954, Ciais and Jouzel, 1994). Air at higher altitudes and latitudes has on average experienced more cooling and rainout and is thus increasingly depleted of heavy isotopes, reflected in negative δ values."

*P2, L31: as long as the air column is unsaturated*

REPLY: An exchange of water molecules occurs in both directions under both saturated and unsaturated conditions. At conditions well below saturation, the exchange is strongly one-sided and the condensation of molecules can be practically neglected. At conditions close to saturation however, condensation cannot be neglected, in particular when the isotopic composition of precipitation and vapour are far from their isotopic equilibrium. Thus, the drop and the surrounding vapour will continuously exchange water molecules, also in unsaturated conditions. We clarified this by adding the following sentence to the end of the paragraph: "It is relevant when the air column is at or near saturation."

*P3, para1: Some introduction about kinetic fractionation would be appropriate here.*

REPLY: Also in response to comments by Reviewer #1 we include now a more detailed introduction of kinetic fractionation by adding:

"In unsaturated conditions, a net transfer of water molecules from the drops to the surrounding air does occur. In addition to the equilibrium fractionation that happens during this transfer, the slower diffusion of the heavy molecules $^{1}H^{2}H^{16}O$ and $^{1}H_{2}^{18}O$ causes additional non-equilibrium or kinetic fractionation. Thereby, lower relative humidity leads to more intense non-equilibrium fractionation. The second-order parameter d-excess (d = $\delta^{2}H-8\cdot\delta^{18}O$) is sensitive to such non-equilibrium conditions, where $^{2}H^{1}H^{16}O$ reaches isotopic equilibrium faster than $^{1}H_{2}^{18}O$. The d-excess quantifies the difference in $^{2}H^{1}H^{16}O$ and $^{1}H_{2}^{18}O$ from their ratio expected during equilibrium conditions as a measure of non-equilibrium (Dansgaard, 1964; Stewart, 1975). Evaporation of rain in unsaturated conditions causes a decrease of d-excess in rain and consequently an increase of d-excess in the surrounding air. Further parameters critically influence this process, such as the drop size distribution (Managave et al., 2016), below-cloud relative humidity (Lee and Fung, 2008), the height of the melting layer, the height of the cloud base (Wang et al., 2016), and vertical wind velocity. Thus, isotopes reflect the conditions acting on rain below the cloud, but in convoluted ways that often render interpretation cumbersome."

**Section 2**
*P4, L2: How much of the data was discarded for both measurements and calibrations? This is typically done to ensure no memory effects on the final values.*

REPLY: We discarded the first 10 minutes of ambient air measurements after each calibration. From the calibration, we discarded the first 5 minutes and the last 30 seconds. Thanks to your comment, we also discovered that the duration of the calibration was in fact 15 min. The text was adjusted accordingly and a sentence to add the discarded amount of data was added:

"The first 5 minutes and last 30 seconds of the calibration, as well as the 10 min ambient air measurements after each calibration were discarded to avoid the influence of memory effect on calibration and the final isotope data."

*P4, L9: Was any oil added to the funnel to prevent evaporation during collection? Did the authors check for potential sample evaporation over 30 minutes, especially if the air was unsaturated? This could especially complicate the interpretation of d-excess values.*

REPLY: We did not add oil to the funnel (or into the vial) or check for potential sample evaporation. We tried to reduce evaporation effects by keeping collection intervals short and by choosing PTFE as material for the funnel, which has reduced adhesion of droplets on its surface compared to other materials.

*P5, L4: This is also complicated by needing to know both temperature of vapor and rain.*

REPLY: We apologize but we did not understand this comment.

*P5, L15: Is ambient temperature accurately reflecting raindrop temperature? Wouldn't it be better to look at "equilibrium precipitation from vapor", since ambient temperature is more likely to represent the ambient vapor? I would suggest repeating the analysis using this direction (or at least check to see if it makes a difference). Especially given the authors' explanation for non-linearity in temperature control of isotope ratios.*

REPLY: Since the net mass flux in an unsaturated environment is from the droplet to the ambient air, we consider it more intuitive to define the equilibrium vapour from precipitation. We emphasize this now when defining this quantity in the manuscript. Ambient temperature is higher than raindrop temperature due to (i) temperature lag of falling precipitation, which carry a bit of the colder temperature aloft and (ii) evaporative cooling. While the temperature difference due to (i) is small (< 0.2°C for d=2mm and lower for smaller drop diameters; Graf 2018; Fig 3,2b between 1000 and 2200m), it is larger due to (ii), and reaches 2.5°C for an ambient relative humidity of 75%. Looking at the 'equilibrium vapour from precipitation' using ambient temperature introduces an error of approximately 2.5‰ for $\Delta\delta$ and 1‰ for $\Delta d$ when assuming a change of the equilibrium difference between vapour and liquid of 1‰/°C and 0.4‰/°C, respectively (cf. Table 1 of the manuscript). Interestingly, the temperature difference between rain and vapour depends mainly on the relative humidity and not on the drop diameter (cf. Fig 3,2b below 1000m in Graf, 2018). The rain temperature can thus be approximated from ambient air temperature and relative humidity measurements instead of measuring it directly.

*P5, L24: Do you get the same results using δ 18O observations instead for eqn (5)?*

REPLY: $\delta^2H$ is more strongly influenced by kinetic fractionation than $\delta^{18}O$. The results, in particular $\Delta d/\Delta\delta$, are thus numerically different, while the conclusions remain unchanged.

**Section 3**

*P7, L19: I assume the correlations reported here are significant with p values < 0.05? Could add 'significant' to line 19. Correlation with h has been seen by others (Crawford et al 2016), especially in unsaturated semi-arid environments. Correlation with rain intensity/amount is traditionally assigned to the classic 'isotopic amount effect'. Some discussion of these would be appropriate here.*

REPLY: Yes, the reported correlations are all significant with p values < 0.05; "significant" was added as suggested. We have split this section in the revised manuscript and in this revised context the discussion of other literature and of the amount effect does not fit well into the discussion.

*P7, L24: Consider: is the 'surrounding vapor' simply the vapor coming down with the rain in a downdraft? In which case I would expect them to be closely correlated. Can you distinguish this from non-downdraft/near-surface surface vapor being influenced by rain?*

REPLY: With 'surrounding vapour', we denote the local near-surface vapour, which is subsequently altered in terms of isotopic composition by the air falling through it. Downdrafts may contribute to the ambient vapour in convective conditions. The frontal passage considered here was however of dominantly stratiform nature, as confirmed by the low variability of the surface vapour isotope composition. We mention the mainly stratiform nature of the frontal precipitation event in the revised manuscript.

**Section 4**

*Fig 3 and P7, L30: how do you distinguish 'rain samples in equilibrium with vapor' – is this the zero line? I don't see samples at 17 UTC near the zero line for Δδ or for Δd at 21 UTC (looks more like 22 UTC). There are several periods where the uncertainty of Δd overlaps the zero line.*

REPLY: With 'rain samples in equilibrium with vapour' we describe samples near the zero line, ideally with an overlap of the uncertainty with the zero line. For Δd, this is the case for several samples, in particular during the post-frontal phase. We gave the samples around 19 UTC and the four samples just before 21 UTC as examples, since they match the zero line almost perfectly. The uncertainty of Δδ is very small and thus only few overlaps occur. We chose 17 UTC as an example, because then Δδ changes from positive to negative. A hypothetical continuous time series would intersect the zero line and be in equilibrium with vapour, although this is not the case for a particular sample. Choosing the time 17 UTC as example is thus somewhat confusing and was changed to 15 UTC. We adapted the text slightly to clarify this:
"Some rain samples are in equilibrium with vapour for $\delta^2 H$ ($\Delta\delta \approx 0‰$; e.g. at 15 UTC), for *d* ($\Delta d \approx 0‰$; at about 19 and before 21 UTC) or for both ($\Delta\delta$ and $\Delta d \approx 0‰$; at 10 UTC)."

*P7, L32-33: More negative values of Δδ are seen before the frontal passage, with Δd trending towards zero. Some explanation of what is going on here? Also, if there is conservation of the depleted isotopic signature from the cloud, I think this should also be reflected in Δd being zero. Any partial/incomplete tendency towards equilibration should be seen in both signatures from the way equations (5) and (6) are set up. If the argument is for 'conservation', i.e. vapor reflecting the original rain signature in equilibrium (possibly because it is associated with a downdraft associated with the frontal passage), this should be captured in both Δδ and Δd.*

REPLY: We assume that this comment refers to the samples in the two hours before the frontal passage (between 17 and 19 UTC), which are already negative for Δδ but not yet close to zero for Δd.

Considering first the second part of the comment, we agree that the influence of the 'cloud signal' and its (partial) conservation is theoretically reflected in both Δδ and Δd. However, the difference of the isotopic composition of precipitation after formation ('cloud signal') and at the surface is much larger for Δδ than for Δd. $\delta_v$ at cloud-level (and thus the initial $\delta_{p,eq}$) is very depleted compared to vapour at the surface. Thus, a large change in $\delta_p$ has to be performed by below-cloud processes in order for rain to arrive at the surface in or close to equilibrium with near-surface vapour. A low melting layer, intense rain or a high below-cloud humidity limit the strength of below-cloud processes and conserves some of the negative Δδ. The situation for Δd is different: The difference between 'cloud signal' and $d_{p,eq}$ at the surface is smaller, mainly due to the absence of a strong vertical gradient of $d_v$, and due to the reasons mentioned in P9,L11-14. It is more easily overcome, even if the time for equilibration is limited. The 'cloud signal' is thus only conserved in cases where equilibration is strongly limited. Consider also, that the 'cloud signal' of d-excess is usually close to, but not necessarily 0‰. A d-excess, which is close to 0‰ or even positive is not necessarily the result of incomplete equilibration and conservation of cloud signal. It is rather an evidence that evaporation, which strongly decreases Δd, was weak. In the revised version of the manuscript, this is now illustrated clearly in simulations with an idealised model of below-cloud processes.

Regarding the first part of the reviewer's comment, a decrease in Δδ is not necessarily linked to an increase of Δd because below-cloud processes have different pathways in the ΔδΔd-space. (i) a change in the degree of equilibration can lead to changes in Δδ without having an effect on Δd, if Δd is already fully equilibrated (at the surface or aloft, before Δd is decreased by evaporation) and (ii) a change in the intensity of evaporation will lead to a large change in Δd, whereas Δδ is affected less strongly. We explain the signal of the samples between 17 and 19 UTC as follows: Δd is low because it reflects the low relative humidity before the frontal passage. The increase in relative humidity should cause an increase of Δd, which is

however partially counteracted by the increasing conservation of a negative Δd from aloft. The decrease of Δδ confirms this increasing conservation of the signal from aloft. It can be explained by the increasing precipitation intensity in this phase, which reduces equilibration. The strong increase in Δd during the frontal passage is then mainly caused by reduced evaporation, which also affects Δδ, but less strongly.

*Fig 4, P8 L6-7: Some discussion of how non-equilibrium kinetic effects would influence the evolution of the Δδ and Δd signals would be appropriate here. Why would the early pre-frontal samples be expected to be around the (0,0) point? Falling through an unsaturated atmosphere at 75-85% humidity, I would expect to see more evaporation and kinetic effects. Fig 4a is easier to interpret, the various transitions during the progression of the rain event are harder to read in Fig 4b.*

REPLY: Kinetic effects during evaporation of rain act in the ΔδΔd-space as a vector towards the bottom right side of the diagram, since it decreases Δd and increases Δδ. The slope of this vector is around −1.3 Δd/Δδ (see comment #3 of referee 1, Figure A) with weak dependencies on remaining fraction, temperature and isotopic composition of rain. Non-equilibrium effects are characterised by a vector that points towards (0,0). The trajectory of rain in the ΔδΔd-space depends on the relative contribution of these two vectors. In low relative humidity conditions, the evaporation vector dominates the overall trajectory (or tendency of the interconnected points in Fig. C with decreasing rain rate), which therefore points towards the bottom right of the diagram. During high relative humidity conditions, the evaporation vector is short and equilibration dominates the overall trajectory, which therefore points towards the origin. The explanation of non-equilibrium effects on the evolution of Δδ and Δd signals has been added in the revised manuscript, due to the inclusion of the idealised simulations with a below-cloud interaction model.

Considering the question about the early pre-frontal samples: Assuming that the said rain samples start with Δd≈0 and fall through an atmosphere with a relative humidity of 80-85% (the samples with RH=75% exhibit a clearly negative Δd), we would indeed expect them to be at more negative Δd. While we cannot provide a final explanation, possible reasons are, e.g., a positive initial Δd, which first has to be compensated by evaporation, before negative Δd occurs, or that the near-surface humidity is not representative of the air column, which may me more saturated in this period of the event.

Lastly, we agree that the various transitions are not perfectly obvious in Fig. 4b, but less obscure than when we connect the dots or display sample numbers. The longer-term progression and transitions should however stand out sufficiently in Fig. 4b to allow following the discussion.

**Section 5**

*P9, para 2: Here the discussion of Δδ and Δd is in terms of precipitation, while previously it was defined as the difference in precipitation-equilibrated and surface vapors.*

REPLY: The use of Δδ and Δd is not necessarily confined to precipitation signals at the surface. It makes sense to apply it to precipitation aloft when studying its evolution in the ΔδΔd-space during the sedimentation from the cloud to the ground. To keep a constant frame of reference with height, we always use surface values of $\delta^2H_v$ and $d_v$, while values from aloft can be used for $\delta^2H_{p,eq}$ and $d_{p,eq}$. Thus, we can use Δδ and Δd to describe the isotopic composition of precipitation and its evolution during the sedimentation from the cloud to the ground, which can help us to understand the processes that determine the location of precipitation at the surface in the ΔδΔd-space. This is now explained in the revised version of the manuscript. More detailed explanations of the evolution of precipitation in the ΔδΔd-space are illustrated in Fig. B (see below), which is also included in the revised manuscript as part of the idealised model simulations. Furthermore, we added clarifying subscripts to eqs. (5) and (6) to denote that the vapour isotope composition is always taken from the surface, while the equilibrium precipitation composition can be taken for any level of the column to describe a temporal evolution:

$$\Delta\delta = \delta^2H_{p,eq} - \delta^2H_v^{sfc} \quad \text{and}$$
$$\Delta d = d_{p,eq} - d_v^{sfc} \ .$$

*P9, L11: Assuming the precipitation is in equilibrium with the vapor it formed from in the cloud (through temperature equilibration), I would expect the Δd of rain versus 'vapor-where-the-rain-formed-in-the-cloud' to be small, but it could still be vastly different from the Δd at the surface.*

REPLY: See also first part of the answer to comment P7 and P9. Intuitively, one would indeed assume, that the difference between $d_{p,eq}$ and $d_v$ (at any given height, i.e. a local value of Δd) is small where precipitation is formed. This is however not the case due to the following reasons:

Kinetic effects during formation through the Wegener-Bergeron-Findeisen process in mixed-phase clouds involves supersaturation and thus kinetic effects, which can increase $d_{p,eq}$ by more than 10‰. Also, the d-excess change during equilibrium fractionation is very different for the vapour-solid than for the vapour-liquid transition. Thus, $d_{p,eq}$ of rain is lower by 13.4‰ compared to $d_{p,eq}$ of snow for $\delta^2H_l = -120$‰ and $d_l = 0$‰ at 0°C. (see Table 1). When changing the phase across the melting layer, $d_{p,eq}$ of precipitation thus decreases. The increase due to kinetic processes and the decrease due to the difference between the vapour-solid and vapour-liquid transition partly compensate each other. In the case presented in Fig. B, the difference between $d_{p,eq}$ and $d_v$ (the local Δd) after formation and at the 0°C-isotherm is negative (–7.5‰ and –5‰; note that the 'column Δd' is shown. The 'local Δd' is however almost equal, since $d_v \approx d_v^{sfc}$). Under different conditions (e.g., lower formation temperatures, which enhance the kinetic effects), the difference might as well be positive.

The locally calculated Δd in the cloud and the Δd at the ground, as defined in the manuscript, can be vastly different due to three reasons: (i) If the local Δd in the cloud is very different from 0‰ and equilibration will reduce its absolute values, (ii) if evaporation strongly decreases $d_{p,eq}$, and (iii) if the vertical gradient of $d_v$ is large. This is underlined by the case presented in Fig. B: The local Δd in the cloud is not small, but the difference between the local Δd in the cloud and on the ground is small (at least for D=0.5 mm and 1 mm).

In addition to the inclusion of the idealised model simulations, we added the vapour-solid transition at 0°C to Table 1 and deleted part of the sentence at P9,L14, because this effect is already included in the definition of $d_{p,eq}$: "and by a decrease due to the non-linearity of the δ-scale (Dütsch et al., 2017)".

*P9, L15: Raindrops will also get more enriched during evaporation into an unsaturated atmosphere, because the lighter isotopes are preferentially evaporated, which will make the ambient vapor more depleted.*

REPLY: We did not discuss the effect of below-cloud processes on ambient vapour, which is of course also something to consider. However, on short enough time-scales (a hydrometeor falling from the cloud to the ground), the effect on vapour can be neglected, since the amount of vapour in an air parcel exceeds the amount of liquid or solid by far, especially for the rain rates we measured (for the calculation see Section 4.2 in Graf, 2018). The effect on vapour would only appear over a longer time period. In the event we present here, a part of the gradual depletion of vapour after ~16 UTC could be caused by interaction with falling precipitation or downward motion of the air, which introduces depleted moisture. This discussion has been included in the description of the time evolution of the vapour isotope measurements of the front.

[Figure]

Figure A: Slope of evaporation line, Δd /Δδ, as a function of (top) temperature and remaining fraction, (middle) initial $\delta^2 H_p$ and remaining fraction, and (bottom) temperature and initial $\delta^2 H_p$.

[Figure]

Figure B: Simulated evolution of the isotope composition of a droplet using the idealised below-cloud process model. (a) Δδ vs. droplet elevation, (b) Δd vs. droplet elevation, (c) Δδ-Δd diagram for the droplets. Coloured lines denote different droplet diameters, black line denotes background vapour composition.

[Figure]

Figure C: Sensitivity of Δδ and Δd for the surface vapour and equilibrium vapour for precipitation to variations of the idealised model setup. For details see the revised manuscript.

---

## Referee Report (RR1)

[referee-annotated manuscript omitted]

---

## Author Response (AR2)

**A new interpretative framework for below-cloud effects on stable water isotopes in vapour and rain**

Authors: Pascal Graf, Heini Wernli, Stephan Pfahl, and Harald Sodemann

We are grateful for the additional comments from reviewer 2 and for his/her general appreciation of the revised version of our paper. Below we provide point-by-point replies (in blue) to the reviewer's comments (in black).

**Reviewer 2**

Abstract
L6: define equilibration or rephrase to 'exchange between ambient air and raindrops'

We add "… both rain evaporation and equilibration (i.e., the exchange of isotopes between raindrops and ambient air)"

Section 3
P9, L13: The similarity between these two quantities could also reflect the fact that the vapor you are measuring is part of a downdraft that came down with the rainfall. In response to my comment on this earlier, the authors said the surface vapor was distinguishable from downdraft vapor because it is a stratiform event. I am not sure this is necessarily true. As pointed out later in this paragraph, mesoscale influence is difficult to distinguish and if the vapor is part of an air mass brought in by a mesoscale system it wouldn't matter if it was a convective or stratiform event for the initial values. I agree that the progression of the isotope values during the rain event will differ depending on whether it is convective or stratiform.

In further clarification, we rephrased this paragraph as follows: "Alternatively, part of the vapor sampled at the surface could have been transported downwards from cloud formation levels by convective downdrafts. In the case analysed here, this influence may be limited due to the mainly stratiform character of the event. Nonetheless, it remains a principal challenge to identify the influence of below-cloud processes in joint observations of vapour and precipitation. One example are signals from meso-scale meteorological processes, such as the transition between airmasses at the weather front."

Section 4
P10, L14: It would be interesting to see if a non-constant isotopic profile could be included in your model. (future work!)

We agree, this would be interesting (and possible with our model), in particular in situations where we have profile observations from observations (e.g., as collected during HyMeX, Sodemann et al. 2017).

P11, L20: Where does this number (63‰) come from?

The main reason for this difference is the depletion of isotopes with height in the background profile resulting from Rayleigh distillation, which contributes 75‰. The exact numerical value is then further influenced by the temperature-dependent fractionation during droplet formation, for

which we assume the Wegener-Bergeron-Findeisen process, and the jump at the 0°C isotherm due to the phase change. We added an explaining sentence to the revised manuscript: "This number stems mainly from the depletion of the background vapor profile (-75‰,). For simplicity, we only show vapor above liquid here, which results in a Δδ of -63‰."

P12, L8: I don't understand which data points from Fig. 3 should line up with Fig. 5c and which points are specifically being referenced in Fig. 5c.

We agree, this sentence was not clear, since Fig. 5c is from idealized simulations. What we wanted to say is "Note that the measured data points of Δδ and Δd shown in Fig. 3 can be compared with the values from our idealized simulations at the final (surface) location shown in Fig. 5c. Therefore, we now display …".

Section 5
P13, L8: P6 L25 says horizontal and vertical air motion are neglected in the model. Is this referring to ongoing future work?

We don't see a contradiction. In P13 L8 we just wanted to list all possible influences, but with our model we address not all of them.

Technical Corrections

Section 4
P10, L18: processed -> processes

Corrected, thank you.

Section 6
P15, L1: imprints 'on'

Corrected, thank you.

P15, L15: 'this type of event' or 'these types of events' (this line used to be in the Abstract I think, I pointed this technical correction out previously)

We think that "this type of events" is correct (and we corrected this in the Abstract).

[revised manuscript text omitted]